# Quantifying nitrogen fixation by heterotrophic bacteria in sinking marine particles

Subhendu Chakraborty [1,2,5✉], Ken H. Andersen [2], André W. Visser [2], Keisuke Inomura [3], Michael J. Follows[4] & Lasse Riemann [1✉]

Nitrogen ($N_2$) fixation by heterotrophic bacteria associated with sinking particles contributes to marine N cycling, but a mechanistic understanding of its regulation and significance are not available. Here we develop a mathematical model for unicellular heterotrophic bacteria growing on sinking marine particles. These bacteria can fix $N_2$ under suitable environmental conditions. We find that the interactive effects of polysaccharide and polypeptide concentrations, sinking speed of particles, and surrounding $O_2$ and $NO_3^-$ concentrations determine the $N_2$ fixation rate inside particles. $N_2$ fixation inside sinking particles is mainly fueled by $SO_4^{2-}$ respiration rather than $NO_3^-$ respiration. Our model suggests that anaerobic processes, including heterotrophic $N_2$ fixation, can take place in anoxic microenvironments inside sinking particles even in fully oxygenated marine waters. The modelled $N_2$ fixation rates are similar to bulk rates measured in the aphotic ocean, and our study consequently suggests that particle-associated heterotrophic $N_2$ fixation contributes significantly to oceanic $N_2$ fixation.

[1] Department of Biology, Marine Biological Section, University of Copenhagen, Helsingør, Denmark. [2] Centre for Ocean Life, DTU Aqua, Technical University of Denmark, Kgs.Lyngby, Denmark. [3] Graduate School of Oceanography, University of Rhode Island, Narragansett, RI, USA. [4] Department of Earth, Atmospheric and Planetary Sciences, MIT, Cambridge, MA, USA. [5] Present address: Systems Ecology Group, Leibniz Centre for Tropical Marine Research (ZMT), Bremen, Germany. ✉email: schakraborty@bio.ku.dk; lriemann@bio.ku.dk

Nitrogen (N) is an essential element for all living organisms but its availability often limits the growth and productivity of terrestrial and aquatic ecosystems. Although molecular dinitrogen gas ($N_2$) is highly abundant in the marine water column, only specific prokaryotes that can fix $N_2$ (diazotrophs) using the nitrogenase enzyme complex[1] assimilate this form of nitrogen. Nevertheless, nitrogen fixation maintains the inventory of biologically available nitrogen in the open oceans, which fuels primary production[1,2], and thereby affects the biogeochemical cycling of both nitrogen and carbon[3].

$N_2$ fixation was thought to exclusively be carried out by cyanobacteria in the oligotrophic and sunlit upper layers of the tropical and subtropical oceans (reviewed in Zehr[4]). However, accumulating evidence shows that $N_2$ fixation is surprisingly widespread, for example in the deep sea[5], nutrient-rich coastal waters[6], and cold Arctic waters[7]. Moreover, analyses of genes (*nifH*) encoding the enzyme complex used for $N_2$ fixation document that non-cyanobacterial diazotrophs are almost ubiquitous across the world's oceans, often dominate *nifH* gene libraries over cyanobacteria, and occasionally express nitrogenase[8–10]. Hence, the emerging picture shows $N_2$ fixation as a global marine process partially carried out by non-cyanobacterial diazotrophs, but their ecology and contribution to total $N_2$ fixation remain enigmatic[10].

Nitrogenase is irreversibly inactivated by $O_2$[11]. Cyanobacteria adopt several strategies to protect nitrogenase from inactivation by $O_2$[12]. Heterotrophic diazotrophs may under rich culture conditions surround cells with extracellular polymers[13] to lower the permeability to extracellular $O_2$ to protect the nitrogenase, but since this is highly energy-demanding[14,15] it is an unlikely strategy in the relatively nutrient-poor marine water column. Recent work, inspired by the pioneering work of Paerl et al.[16,17], has suggested that heterotrophic $N_2$ fixation takes place in low-oxygen or anaerobic microzones associated with marine particles (reviewed in Riemann et al.[18] and in Bombar et al.[10]). Indeed, anaerobic microzones are occasionally associated with marine particles[19,20]. Recent studies show $N_2$ fixation is stimulated by the presence of particles[21,22] and that heterotrophic diazotrophs are associated with plankton specimens[23,24] and marine aggregates[25,26]. Hence, beyond doubt, marine particles provide, at least ephemeral, conditions suitable for $N_2$ fixation by heterotrophic bacteria.

Cellular $O_2$ removal by diazotrophs is considered highly energy-demanding, even more energetically expensive than $N_2$ fixation per se[14]. Hence, considerable amounts of labile carbon (e.g., carbohydrate and amino acids) are required to sustain particle-associated microbial respiration beyond the specific energy requirements for diazotrophy. Interestingly, preferential microbial utilization of N-rich organics on particles[27] and release of $NH_4^+$ [28] may increase particle C:N ratios over time[27], gradually making N acquisition by $N_2$ fixation increasingly advantageous.

While synthesizing ATP during respiration, $O_2$ is used as the most common and favorable form of electron acceptor by prokaryotes. In the absence of $O_2$, other electron acceptors (e.g., $NO_3^-$ and $SO_4^{2-}$) may be used in a stepwise manner according to their free energy yields[29], with a rather small drop-off in the theoretical energy yield for $NO_3^-$ respiration, followed by $SO_4^{2-}$ respiration with almost tenfold less energy yield per electron donor[30]. However, $SO_4^{2-}$ respiration is likely the primary form of anaerobic respiration supporting $N_2$ fixation since $SO_4^{2-}$ reducing diazotrophs have been widely found[31,32], also on marine particles[25].

Another factor likely regulating $N_2$ fixation in sinking marine particles is the particle size, which varies from micrometers to several millimeters[33]. The particle size spectrum follows a power law relationship showing a decrease in particle abundance with increasing size[34]. Because of smaller surface-to-volume ratios, large particles are more likely to develop an anoxic interior suitable for $N_2$ fixation. Moreover, particles face changing $O_2$ and $NO_3^-$ concentrations while descending in the water column. The rate of change depends on particle sinking speed, but no universal size-sinking speed relationship exists[35]. Although all these external factors can have huge influences, the extent by which they affect heterotrophic $N_2$ fixation inside sinking particles is currently unclear.

To quantitatively analyze the conditions when heterotrophic $N_2$ fixation occurs on sinking particles, we present a trait-based model of heterotrophic bacteria associated with sinking particles. This effort aims to encapsulate an understanding of the dynamics between (micro)environmental conditions and the requirements and constraints of heterotrophic $N_2$ fixation. Specifically, the model captures basic cellular processes determining growth and $N_2$ fixation in an individual cell and then scales up to the population level to address particle dynamics and the contribution to total $N_2$ fixation in the water column. We also examine how the size of particles, initial concentrations of polysaccharide and polypeptide, and environmental $O_2$ concentration influence heterotrophic $N_2$ fixation inside sinking particles, and the succession of aerobic and anaerobic respiration as support for $N_2$ fixation. In doing so we identify potentially testable hypothetical consequences: (H1) excess acquired N released by cells and hydrolysis products diffuse away from the particle and contribute to an organic solute trail in the water column as the particle sinks. (H2) $N_2$ fixation by heterotrophic diazotrophs depends on the generation of particle-associated low-oxygen microenvironments. (H3) During the "life span" of a sinking marine particle there is an ephemeral window of opportunity where environmental conditions are conducive for heterotrophic $N_2$ fixation. (H4) $SO_4^{2-}$ reduction is more important for $N_2$ fixation within sinking particles than $NO_3^-$ reduction. (H5) The particle sinking speed and concentrations of $O_2$ and $NO_3^-$ in the water column affects $N_2$ fixation rates. Although the model is developed to investigate $N_2$ fixation, it also provides critical insights on biochemistry and microbial respiratory processes inside sinking particles.

## Results and discussion

**Overview of the model.** The overall model consists of a "cell model" and a "particle model". The cell model describes basic cellular processes, like uptake of resources, respiration, growth, and $N_2$ fixation rate. The cell model is embedded in a dynamic model, called the particle model, that deals with interactions of cells with the available abiotic factors (polysaccharide, polypeptide, $O_2$, $NO_3^-$, $SO_4^{2-}$) over time.

The cell model describes a population of facultative $N_2$ fixing heterotrophic bacteria growing inside a particle sinking through a water column. A schematic representation of the processes inside a single cell is presented in Fig. 1 and the full description of mathematical forms and equations are provided in the "Methods" section. The cell uses ectoenzymes to degrade polymers (polysaccharides and polypeptides) to oligomers or monomers (glucose and amino acids) that it can efficiently take up to fulfill its C and N requirements. The uptake of glucose and amino acids follows Michaelis–Menten kinetics. The model accounts for acquired C and N to ensure that the cell satisfies its needs for both. While glucose uptake provides only C, amino acids provide both C and N (Eqs. 8 and 9).

C obtained from glucose and amino acids is respired to carry out resource uptake, cellular maintenance (Fig. 1), and standard metabolism (Eq. 13). $O_2$, $NO_3^-$, and $SO_4^{2-}$ are used as electron acceptors in a stepwise manner in order of their free energy yield to perform respiration[36]. In the absence of sufficient $O_2$, the cell

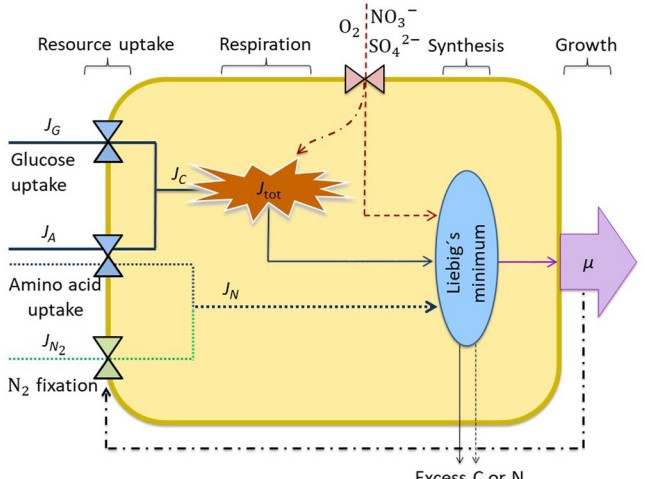

**Fig. 1 Schematic representation of the cellular processes.** It shows how fluxes of carbon (C), $J_C$ (solid lines), fluxes of nitrogen (N), $J_N$ (dotted lines), and electron acceptors ($O_2$, $NO_3^-$, and $SO_4^{2-}$; red dashed line) are combined (blue ellipse) to determine growth rate after paying the cost of respiration (brown explosion). Triangle symbols represent the functional responses for the uptake mechanisms and diffusive inflow of $O_2$, $NO_3^-$, and $SO_4^{2-}$. $J_{tot}$ represents respiration that includes costs of uptake and mobilization of resources for synthesis, construction/maintenance of structure, and ectoenzyme production. The ellipse represents the synthesis of biomass from the available C, N, and electron acceptors following Liebig's law of the minimum. Any excess assimilated C or N is excreted from the cell. $\mu$ represents the division rate. The black dashed-dotted line represents the regulation of $N_2$ fixation to optimize growth rate and the red dashed-dot line represents the regulation of respiration by electron acceptors.

uses $NO_3^-$ to continue respiration, although all N necessary for growth comes from organic sources and $N_2$ fixation (whenever possible). The further need of an electron acceptor is fulfilled by $SO_4^{2-}$.

The cell can carry out $N_2$ fixation to supplement its N requirement. It regulates the rate of $N_2$ fixation to optimize its growth rate. As nitrogenase is irreversibly inhibited by $O_2$[11], the cell needs low $O_2$ conditions inside particles or increased respiration to make the cell $O_2$ free and thereby enable $N_2$ fixation.

The synthesis of biomass using available C and N from resource uptake and electron acceptors follows Liebig's law of the minimum and is constrained by the cellular C:N ratio ($\rho_{CN,B}$) (Eq. 28). Any excess assimilated C or N is excreted from the cell. The cell division rate $\mu$ is found from the mass-specific synthesis rate.

The particle model consists of a sinking particle that contains polysaccharides and polypeptides and is colonized by facultative nitrogen-fixing bacteria (Supplementary Fig. S1). Only fractions of these polymers are considered labile, i.e., accessible by bacteria. Bacterial enzymatic hydrolysis converts labile polysaccharides and polypeptides into monosaccharides (glucose) and amino acids that are efficiently taken up by bacteria. Excess glucose and amino acids diffuse out of the particle to the surrounding environment, while $O_2$ and $NO_3^-$ diffuse into the particle from the surrounding water. $SO_4^{2-}$ is the second most abundant anion in seawater with an estimated concentration of 28 mM[37] (roughly three orders of magnitude higher concentration than $NO_3^-$) whereas $N_2$ is also plentiful with an average concentration of 0.4 mM in seawater[37] (more than two orders of magnitude higher concentration than $NO_3^-$). Due to these high concentrations of

$SO_4^{2-}$ and $N_2$ inside particles, the uptake is assumed to be limited by the cellular maximum uptake capacities and not by the rate of diffusion toward the cell. Depending on the available concentrations of glucose, amino acids, $O_2$, and $NO_3^-$ inside the particle, bacteria carry out $N_2$ fixation (Eq. 30). Fe, an essential component in the nitrogenase complex, is considered nonlimiting as sinking particles contain high levels of Fe[38]. The predation on bacteria is represented by a linear mortality term. The interactions between particle, cells, and the surrounding environment are explained in the supplementary Fig. S1, equations are provided in Table 1, and a full description of the particle model is provided in the "Methods" section.

**Biochemical dynamics inside a particle under static environmental conditions.** The dynamics inside a particle of radius 0.125 cm with initial polysaccharide and polypeptide concentrations of $2.6 \times 10^8$ µg G $L^{-1}$ and $1.6 \times 10^8$ µg A $L^{-1}$, with relative lability of 0.238 and 0.5, respectively, are depicted in Fig. 2. We simulate a population of bacterial cells of radius 0.29 µm (50 fg C cell$^{-1}$) growing inside a particle where the surrounding glucose, amino acids, $O_2$, $NO_3^-$, and $SO_4^{2-}$ concentrations are kept fixed at 50 µg G $L^{-1}$, 5 µg A $L^{-1}$, 50 µmol $O_2$ $L^{-1}$, 15 µmol $NO_3$ $L^{-1}$, and $29 \times 10^3$ µmol $SO_4$ $L^{-1}$. Here, concentrations inside the particle are given as per liter of particle and outside as per liter of water. A full description of the included parameters and their values is available in Supplementary Material S1 and Table S1. The bacteria hydrolyze labile polysaccharides and polypeptides into glucose and amino acids using ectoenzymes (Fig. 2a, b). As a result, glucose and amino acid concentrations increase inside particles (Fig. 2c, d), which causes a high growth rate of cells (~3.6 d$^{-1}$; Fig. 2j), an increase in bacterial abundance (Fig. 2g), and a decrease in labile polysaccharide and polypeptide concentrations. The occurrence of such a high bacterial abundance is not rare inside natural sinking particles[26,39,40]. The growth rates observed in the model are high for the temperature regime but conceivable given that there are some reports of extremely high growth rates in particle-associated bacteria albeit at higher temperatures and in different environments than modeled here[41,42]. The increased community respiration (Fig. 2h) decreases $O_2$ concentration and eventually leads to anoxia in the particle interior (Fig. 2e). This is consistent with ephemeral anoxia inside marine aggregates[20] and the anoxia observed inside suspended cyanobacterial colonies of comparable size[19]. The gradual formation of low-oxygen or anoxic conditions and depletion of organic N (amino

**Table 1 Equations for the particle model.**

| Variables | Equations | |
| --- | --- | --- |
| Bacteria (cells $L^{-1}$) | $\frac{\partial B}{\partial t} = \mu^*(G, A, X_{O_2}, X_{NO_3})B - m_B B$ | 1 |
| Labile polysaccharides (µg G $L^{-1}$) | $\frac{\partial C_L}{\partial t} = -J_C B$ | 2 |
| Labile polypeptides (µg A $L^{-1}$) | $\frac{\partial P_L}{\partial t} = -J_P B$ | 3 |
| Glucose (µg G $L^{-1}$) | $\frac{\partial G}{\partial t} = J_C B - J_G B + D_M \left( \frac{\partial^2 G}{\partial r^2} + \frac{2}{r} \frac{\partial G}{\partial r} \right)$ | 4 |
| Amino acids (µg A $L^{-1}$) | $\frac{\partial A}{\partial t} = J_P B - J_A B + D_M \left( \frac{\partial^2 A}{\partial r^2} + \frac{2}{r} \frac{\partial A}{\partial r} \right)$ | 5 |
| Oxygen (µmol $O_2$ $L^{-1}$) | $\frac{\partial X_{O_2}}{\partial t} = -F_{O_2} B + \bar{D}_{O_2} \left( \frac{\partial^2 X_{O_2}}{\partial r^2} + \frac{2}{r} \frac{\partial X_{O_2}}{\partial r} \right)$ | 6 |
| Nitrate (µmol $NO_3$ $L^{-1}$) | $\frac{\partial X_{NO_3}}{\partial t} = -J_{NO_3} B + \bar{D}_{NO_3} \left( \frac{\partial^2 X_{NO_3}}{\partial r^2} + \frac{2}{r} \frac{\partial X_{NO_3}}{\partial r} \right)$ | 7 |

All quantities vary with time $t$ and with distance from the center $r$. The operator in the brackets represents diffusion in spherical coordinates. Definitions, units, and values of each of the parameters are provided in Supplementary Table S1.

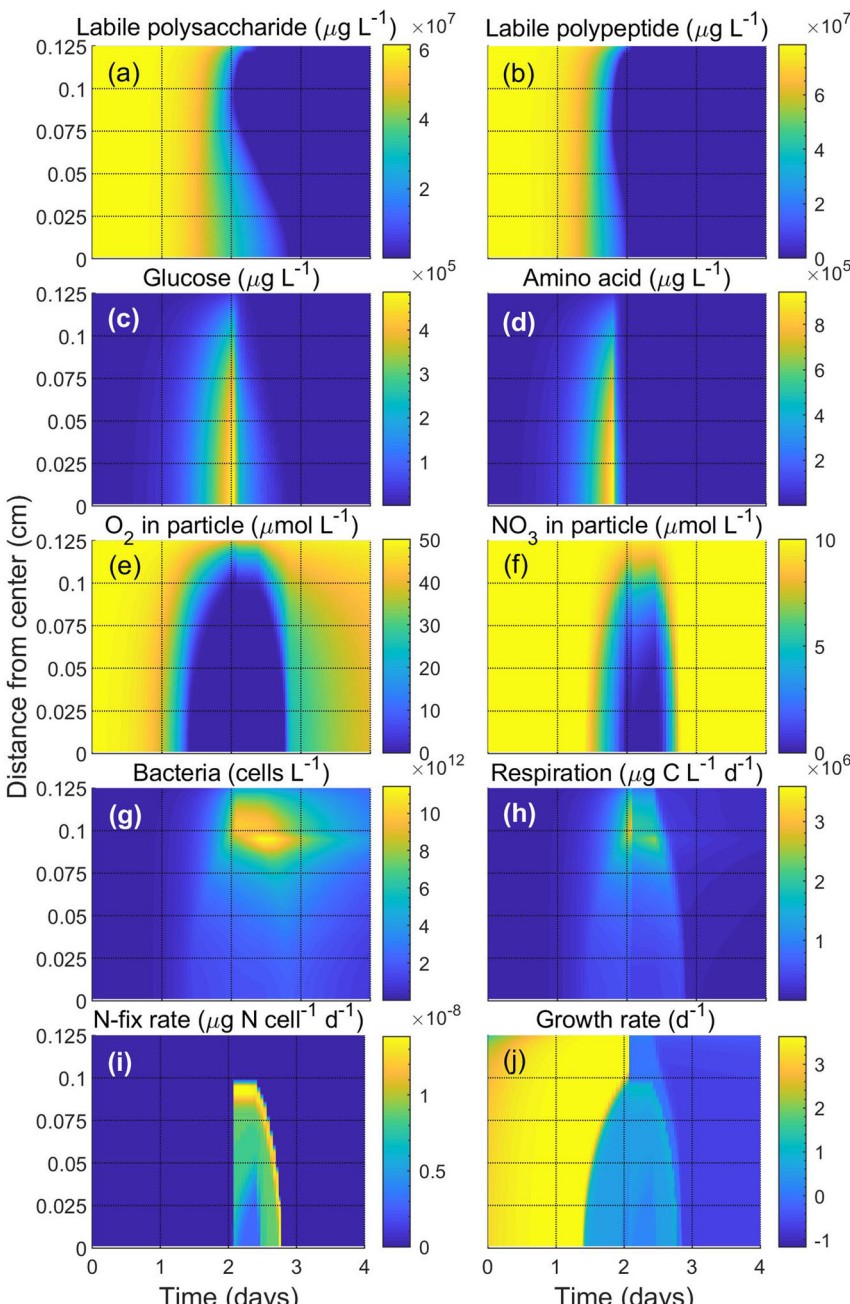

**Fig. 2 Dynamics inside a particle of radius 0.125 cm with time. a** Labile carbohydrate, **b** labile polypeptide, **c** glucose, **d** amino acid, **e** $O_2$ in particle, **f** $NO_3^-$ in particle, **g** bacterial abundance, **h** respiration rate, **i** $N_2$ fixation rate, and **j** growth rate are shown along the particle radius over time. Parameters and concentrations of surrounding factors are taken from Table S1.

acids) facilitates $N_2$ fixation (Fig. 2i), supported by aerobic respiration followed by $NO_3^-$ respiration (Fig. 2e, f). The lesser energetic yield of $NO_3^-$ respiration leads to a reduced growth rate (~0.6 d$^{-1}$). Furthermore, when $NO_3^-$ becomes exhausted (Fig. 2f), cells respire $SO_4^{2-}$ (not shown in the figure) and the growth rate becomes very low (~0.2 d$^{-1}$). Because of the energetic constraints, $N_2$ fixation during this phase also becomes low. The presence of such $NO_3^-$ and $SO_4^{2-}$ reducing bacteria is common in sinking particles[43]. Eventually, $N_2$ fixation ceases due to increased $O_2$ levels as the exhaustion of labile carbon in the particle decreases cell concentration and $O_2$ influx exceeds $O_2$ consumption (aerobic respiration).

We performed a sensitivity analysis, adapted from earlier studies[44], to investigate how different factors affect the $N_2$ fixation rate inside particles (Supplementary Fig. S2). $N_2$ fixation

rate was found to increase with increased maximum glucose uptake rate and maximum amino acid uptake rate, and decrease with the cost of amino acids uptake, hydrolysis rate of polysaccharide, and the fraction of $O_2$ diffusivity within particles compared to water. Significant decreases in $N_2$ fixation rate were observed with an increase in hydrolysis rate of polysaccharide and a decrease in maximum glucose uptake rate. Not surprisingly, the sensitivity analysis suggests hydrolysis and uptake as key parameters affecting $N_2$ fixation, pinpointing the importance of particle composition and bioavailability for particle-associated $N_2$ fixation.

**Cellular mechanisms of $N_2$ fixation.** To explore the cellular mechanism of $N_2$ fixation, we examine concentrations and rates

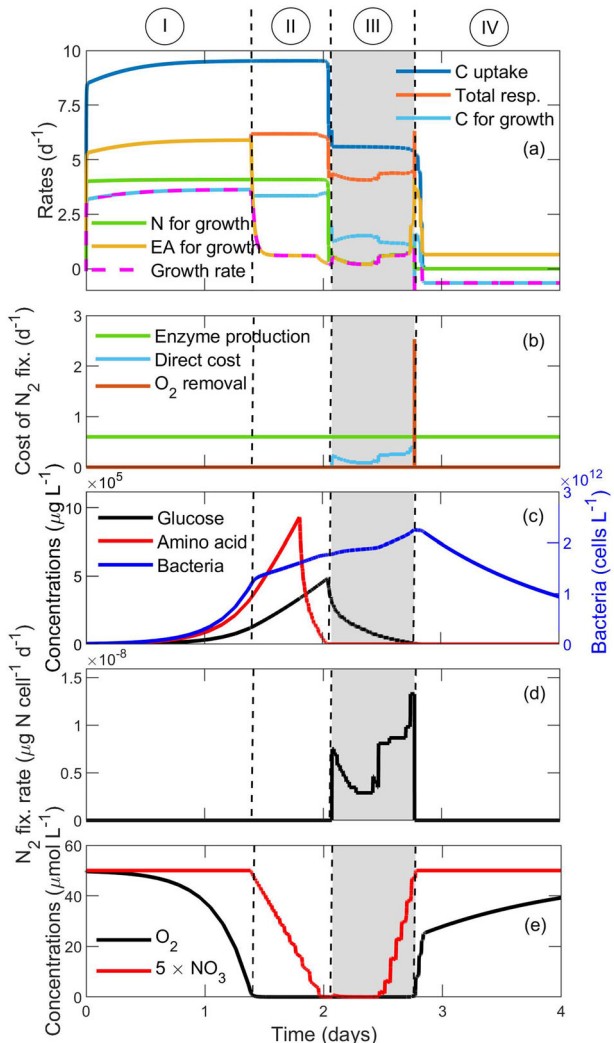

**Fig. 3 Cellular rates and resource concentrations at a radial distance of 0.027 cm from the center of a particle of radius 0.125 cm. a** Size-specific rates of C uptake (dark blue), total respiration (orange), available C for growth (light blue), available N for growth (green), available EA (electron acceptor $O_2$, $NO_3^-$, and $SO_4^{2-}$; yellow), and growth rate of a cell (dashed magenta). Regions I, II, III, and IV represent situations when a cell is limited by C, EA, co-limited by N and EA, N, and showing negative growth rate (see text). **b** Respiratory costs related to $N_2$ fixation in terms of direct respiration, enzyme production, and $O_2$ removal. **c** Glucose, amino acid, and bacterial concentrations in the particle. **d** Cellular $N_2$ fixation rate. **e** $O_2$ and $NO_3^-$ concentrations in the particle. The gray area represents the time interval of $N_2$ fixation.

over time at a radial distance of 0.027 cm from the particle center (Fig. 3). We identify four phases: (I) C limited phase, (II) high respiration phase, (III) $N_2$ fixing phase, and (IV) fading phase. The different phases are based on limitations of either C or N or electron acceptor where the growth rate is determined by the minimum availability of these three substances (Eqs. 28 and 29; Fig. 3a). Available C for growth is the C remaining from total C uptake after paying the respiratory costs, whereas the N available for growth comes from uptake and $N_2$ fixation. A stepwise conceptual flow chart of how different factors are responsible for different events inside particles is provided in Supplementary Fig. S3.

I. Due to high respiratory cost, cells are limited by C, which is seen by the light blue line coinciding with the magenta line

in Fig. 3a. Growth rates are high, up to 3.6 day$^{-1}$. Excess N and hydrolysis products not taken up by cells[27] will diffuse away from the particle and contribute to an organic solute trail in the water column as the particle sinks[45], supporting our hypothesis H1.

II. The large bacterial population causes high community respiration, matching or exceeding the diffusive influx of $O_2$, and anoxia forms in the particle interior (Fig. 3e). Cells start respiring $NO_3^-$ and even $SO_4^{2-}$ reduction happens at the end of this phase (not shown in the figure) when $NO_3^-$ is depleted. Now growth is limited by the availability of electron acceptor (the yellow line coincides with the magenta in Fig. 3a). An organic solute trail rich in both C and N is predicted during this phase. The amino acid concentration decreases rapidly during the final part of this phase (Fig. 3c).

III. Because of our initial choice of polysaccharide and polypeptide concentrations, amino acids are exhausted. In real life, bacterial preferential degradation of N-rich organics results in similar early exhaustion of amino acids[27]. Glucose remains available as C source (Fig. 3c), so cells start fixing $N_2$ to maintain growth (Fig. 3d). However, since the available $O_2$ in the particle is insufficient to support respiration, $NO_3^-$ and $SO_4^{2-}$ also act as electron acceptors during this phase. Cells become co-limited by N from $N_2$ fixation and electron acceptor (green and yellow lines coincide; Fig. 3a) and we predict that the expected solute trail consists only of C during this phase. Because of the lower free energy yield, $N_2$ fixation decreases when using $SO_4^{2-}$ as an additional electron acceptor. Toward the end of this phase, the respiratory cost of glucose uptake decreases with the decrease in glucose concentration to such an extent that there is excess $O_2$ after performing respiration. At that point, cells increase respiration to burn excess $O_2$ to perform $N_2$ fixation for a very short interval of time, resulting in a peak in $N_2$ fixation rate (Fig. 3d) and the respiratory cost for $O_2$ removal (Fig. 3b).

IV. Cells have insufficient C to deal with excess $O_2$ and consequently stop $N_2$ fixation. N becomes the limiting factor for cell growth and growth ceases. Later, the growth rate becomes negative as there is no glucose left needed for basal respiration. Throughout this phase, the bacterial concentration decreases (Fig. 3c).

The identification of different phases yields insights relevant for several of our hypotheses. As the particle sinks, it will be tailed by an organic matter solute trail. Its composition depends on the internal state of the particle. During the "C limited phase" and "High respiration phase", the expected trail consists of both C and N, whereas during the "$N_2$ fixation phase", the trail contains only C. The amount of C and N in the trail can be estimated from the outgoing flux of the organic matter from the particle, supporting our hypothesis H1. Comparing different respiratory costs related to $N_2$ fixation in terms of direct respiration, enzyme production, and $O_2$ removal, it becomes evident that when cells use respiratory protection to keep nitrogenase viable and enable $N_2$ fixation, the related cost becomes much higher than the direct cost for $N_2$ fixation (Fig. 3b). A similar high cost of $O_2$ management during $N_2$ fixation was previously shown to exceed the costs of $N_2$ fixation per se for the heterotrophic soil bacterium *Azotobacter vinelandii*[14]. We therefore conclude that active $O_2$ management by particle-associated heterotrophic diazotrophs is not prevalent, but that they rather depend on the generation of low-oxygen microenvironments by community respiration, supporting our initial hypothesis H2.

The observed fast transitions of glucose and amino acids happen when the exponential growth of bacteria leads to high

bacterial abundance and high degradation of polymers (Figs. 2 and 3c). Similar fast changes in bacterial abundance and co-occurring resource utilization have been observed in an experiment showing degradation of transparent exopolymer particles derived from cultures of the coccolithophore *Emiliania huxleyi*[46]. However, a very sharp transition has been observed for $N_2$ fixation rate (Figs. 2i and 3d). Because of the cost related to $N_2$ fixation, $N_2$ fixation starts only when there is no N available from amino acid (Fig. 3c). The rate of $N_2$ fixation is determined by the available electron acceptors (Fig. 3a) and results in a sharp transition at the beginning of $N_2$ fixation. $N_2$ fixation stops when the C available from glucose is not sufficient to burn excess $O_2$ to keep the cell $O_2$ free; consequently, $N_2$ fixation terminates abruptly.

**Effects of particle size, $O_2$, and initial polysaccharide and polypeptide concentrations**. Marine particles are highly variable in size[33] and chemical composition. For example, the C:N ratio (and implied polysaccharide:polypeptide availability) varies considerably, depending on environmental conditions[47]. Moreover, while descending in the water column, particles face a range of surrounding $O_2$ concentrations. Therefore, we examined the implications for $N_2$ fixation in particles of different sizes under different polysaccharide, polypeptide, and $O_2$ concentrations (Fig. 4). We considered open ocean particles with radius 5 μm to 0.25 cm[48] and estimated the total amount of fixed $N_2$ per particle by allowing bacteria to grow inside particles for 20 days. As expected, large particles provide a suitable environment for $N_2$ fixation. The minimum size of particles where $N_2$ fixation is possible increases with polysaccharide concentration (Fig. 4a), decreases with environmental $O_2$ concentration (Fig. 4c), and attains a maximum at intermediate polypeptide concentration (Fig. 4b). $N_2$ fixation does not occur in particles with radius below ~0.03 cm. The increase in $N_2$ fixation with polysaccharide concentrations is consistent with observations of stimulated $N_2$ fixation in seawater upon the addition of C substrate (e.g., Rahav et al.[22]). Interestingly, maximum $N_2$ fixation occurs at intermediate polypeptide concentrations ($\sim5\times10^7$ μg L$^{-1}$) in large particles (Fig. 4b). Our interpretation is that low polypeptide concentrations do not allow cells to grow to high concentrations and create an anoxic interior, whereas, at high concentrations, cells cover their N demand by amino acid assimilation and, therefore, refrain from $N_2$ fixation.

Maximum $N_2$ fixation in large particles occurs under very low ($\sim0.3$ μmol L$^{-1}$) and intermediate ($\sim80$ μmol L$^{-1}$) $O_2$ levels (Fig. 4c). Under very low $O_2$ concentrations, cellular respiration is expected to occur using $NO_3^-$ and $SO_4^{2-}$ as electron acceptors causing low cellular growth rate and a slowly increasing cell concentration. However, $N_2$ fixation occurs for a long time period and makes the total $N_2$ fixation per particle relatively high (Supplementary Fig. S4d). At intermediate $O_2$ levels, cells are not limited by $O_2$ and reach a high growth rate during the initial phase. Therefore a high cell concentration is rapidly obtained (Supplementary Fig. S4c) causing reduced $O_2$ levels suitable for $N_2$ fixation for a relatively shorter time interval (Supplementary Fig. S4d). The combination of high cellular $N_2$ fixation rate and high cell concentration results in high total $N_2$ fixation per particle. At a high $O_2$ level ($\sim200$ μmol L$^{-1}$), this low $O_2$ period is very short, possibly because of a large diffusion loss of glucose associated with extensive polysaccharide hydrolysis caused by the high cell concentration (Supplementary Fig. S4c). This lowers the total amount of $N_2$ fixed per particle. However, the concentrations of $O_2$, where these two maxima occur, depend on the initial polysaccharide and polypeptide concentrations. With a decrease in these concentrations, the intermediate $O_2$ concentration, where

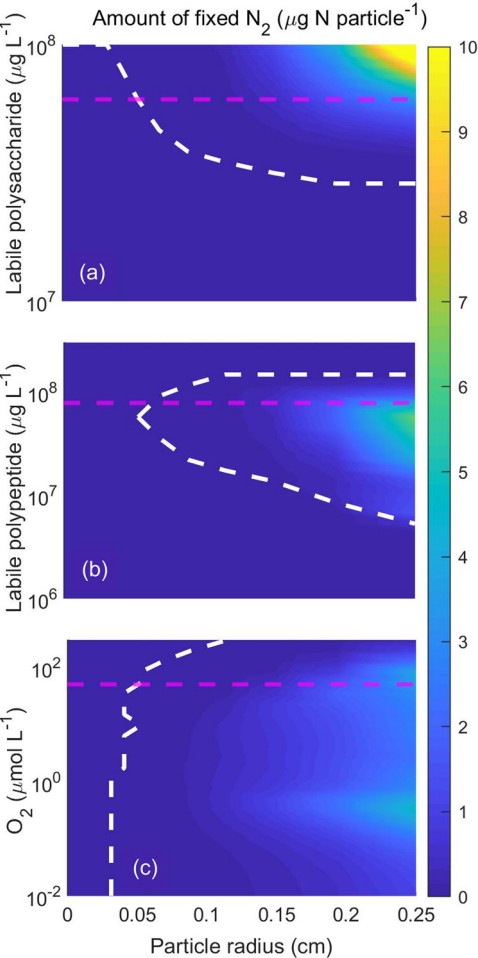

**Fig. 4 Effects of particle size, $O_2$, and initial polysaccharide and polypeptide concentrations on $N_2$ fixation rates.** The total amount of fixed $N_2$ per particle as a function of particle radius and initial labile polysaccharide (**a**), initial labile polypeptide (**b**), and surrounding $O_2$ concentration (**c**). The white dashed line separates regions of occurrence and non-occurrence of $N_2$ fixation ($N_{fix}>10^{-3}$ μg N particle$^{-1}$). The horizontal magenta dashed lines indicate the base value for other plots, e.g., in (**a**), magenta dashed line represents the level of polysaccharide concentration in **b** and **c**. The chosen concentration ranges corresponds to those in natural particles[60,81].

the maximum occurs, decreases (Supplementary Fig. S5) and finally merges with the other maximum at low $O_2$ (not shown). Indeed, empirical results confirm the existence of such optimal $O_2$ concentration for $N_2$ fixation and a level of 6 μmol L$^{-1}$ has been observed for heterotrophic diazotrophs[49].

By simultaneously varying initial labile polysaccharide and polypeptide concentrations in small and large particles at different surrounding $O_2$ concentrations, it appears that $N_2$ fixation is restricted to large particles with high initial polysaccharide and polypeptide concentrations when $O_2$ concentration is high (Fig. 5a). However, under low $O_2$ concentrations, $N_2$ fixation occurs even at lower concentrations of polysaccharides and polypeptides (Fig. 5b). $N_2$ fixation can also occur in relatively smaller particles, however, only when the initial polysaccharide concentration is high and the surrounding $O_2$ concentration is low (Fig. 5c).

The amount of POC present in particles can be considered a proxy for polysaccharide and polypeptide concentrations. The presence of high POC in freshly formed particles from

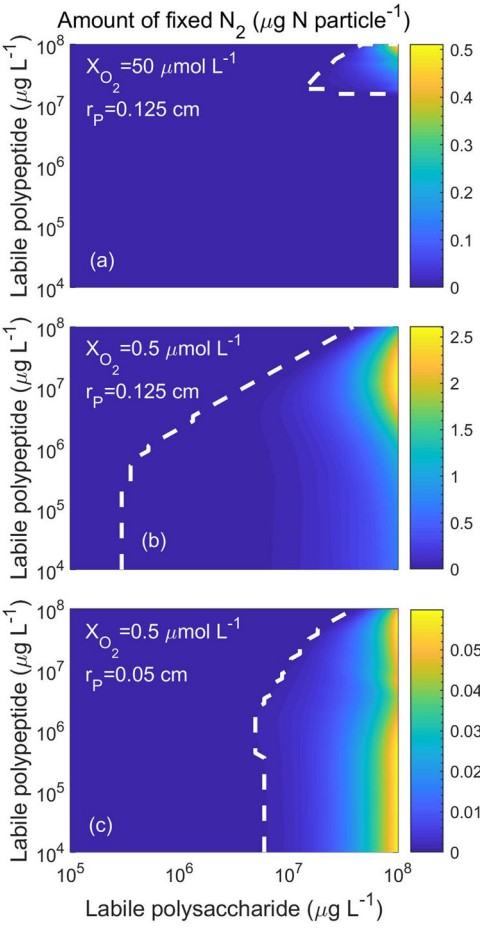

**Fig. 5 The total amount of fixed N₂ at different initial labile polysaccharide and polypeptide concentrations.** N₂ fixation is observed in a large particle of radius 0.125 cm and surrounding O₂ concentrations **a** 50 μmol O₂ L⁻¹ and **b** 0.5 μmol O₂ L⁻¹, and **c** in a relatively smaller particle of radius 0.05 cm and surrounding O₂ concentrations 0.5 μmol O₂ L⁻¹. Line types are similar as in Fig. 4. Note the different ranges of the scale of N₂ fixation.

dense phytoplankton blooms[50] and fecal pellets[51] would increase the likelihood of particle-associated N₂ fixation. Likewise, the seasonally high POC content of particles during late spring in high latitude areas[52] could increase the likelihood of particle-associated N₂ fixation at this time. On the other hand, the size of particles, which also has a profound impact on N₂ fixation, is closely associated with the species responsible for particle formation. For instance, since particles are larger during diatom blooms compared to cyanobacterial blooms[53], an increased likelihood of N₂ fixation during diatom blooms would be expected. The latitudinal variation in particle size spectrum with a dominance of larger particles at higher latitudes compared to smaller particles in the oligotrophic subtropical gyres[54] indicates a greater opportunity for particle-associated N₂ fixation at high latitudes. Hence, the potential for particle-associated N₂ fixation is highly dependent on local dynamics in the size and composition of particles, together with the local O₂ conditions.

**N₂ fixation in sinking particles.** To explore the dynamics of N₂ fixation in sinking particles of different sizes, we use vertical profiles of O₂ and NO₃⁻ in the upper 500 m of the Mauritanian upwelling zone in the North Atlantic Ocean (NAO; Fig. 6)[55]. O₂ concentration drops to hypoxic levels (~62.5 to 157 μmol L⁻¹) in

the water column between 100 and 600 m (Fig. 6a). N₂ fixation coincides mainly with the presence of an anoxic particle interior. The amount of fixed N₂ increases with particle size and because of higher sinking rates and more C to fuel respiration, the existence of both anoxic interior (regions within magenta lines) and N₂ fixation (within white lines) in large particles occur in deeper waters and persist for longer time (Fig. 6b). We presume that N₂ fixation stops in deep water when labile material in the particle is exhausted. This shows that the window of opportunity where environmental conditions are conducive for heterotrophic N₂ fixation is ephemeral, supporting our hypothesis H3.

In the anoxic particle interior, NO₃⁻ and SO₄²⁻ function as electron acceptors. The model suggests that the fraction of particle volume where denitrification occurs (maximum 6%) is much smaller than the fraction of volume of occurrence of SO₄²⁻ reduction (maximum 90%) (Fig. 6c, d). The model predicts that despite being energetically profitable, NO₃⁻ does not play a big role in N₂ fixation and SO₄²⁻ reduction appears as the key anaerobic process within sinking particles. This is due to very high cell concentration near the surface of the particle (Fig. 2g) that creates a high respiratory demand for electron acceptors, exhausts NO₃⁻ close to the particle surface, and prevents NO₃⁻ from reaching the particle interior (Fig. 2f). As a result, N₂ fixation in most of the particle interior is supported by SO₄²⁻ respiration, which confirms the importance of SO₄²⁻ reduction in particle-associated N₂ fixation compared to NO₃⁻ reduction, supporting hypothesis H4. Diazotrophy among SO₄²⁻ reducing bacteria is well established in various marine environments[56,57].

**Influence of sinking speed on particle-associated N₂ fixation.** The speed at which particles sink is a critical parameter since it determines the duration of exposure to environmental conditions (e.g., O₂) that influence N₂ fixation inside particles. Sinking speed is affected by a multitude of factors related to particle composition, size, and density[58–60], and no universal size-sinking velocity relationship exists[35]. We, therefore, examine scenarios with three types of particles: (1) natural marine snow measured in situ off California[61], (2) laboratory-made diatom aggregates, and (3) coccolithophore aggregates measured in vitro[50]. Sinking speed is lowest for natural marine snow followed by diatom aggregates and coccolithophore aggregates (Fig. 7a). For the sake of simplicity, these particles are assumed to vary only in their sinking speeds and not in their initial concentrations and lability of polysaccharides and polypeptides. We again use the vertical profile of O₂ and NO₃⁻ at the NAO but extended to 1500 m depth with a hypoxic region between 100 and 600 m depth (Fig. 7b).

Three different aspects are evident from the analysis. First, when particles sink at a speed similar to natural marine snow (~15-75 m d⁻¹), anoxic microenvironments are created (regions within magenta lines) and N₂ fixation happens (indicated by color) in particles within the hypoxic zone (Fig. 7e). However, with higher sinking speeds similar to diatom (~2 to 180 m d⁻¹) and coccolithophore (~20 to 375 m d⁻¹) aggregates, the existence of anoxia and N₂ fixation inside particles can extend beyond the hypoxic strata of the water column (Fig. 7d, e); both anoxic interior and N₂ fixation are predicted even at 1800 m depth for large coccolithophore aggregates (not in the figure). Second, the depth window where N₂ fixation occurs increases with particle sinking speed (Fig. 7c–e). Finally, our study predicts that the highest N₂ fixation rate (1.46 μg N (cm³ particle)⁻¹ d⁻¹) is attained in large particles at intermediate sinking velocities, similar to that of diatom aggregates (Fig. 7d). Since the presence of higher concentrations of NO₃⁻ can stimulate NO₃⁻ respiration in a relatively larger volume fraction of particles, and due to

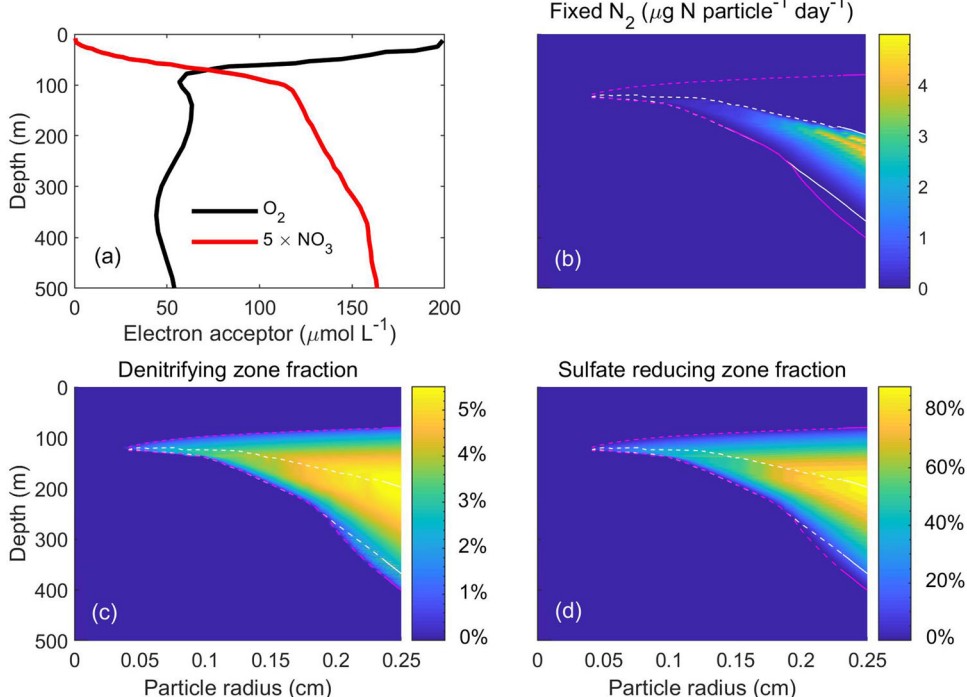

**Fig. 6 Dynamics inside sinking particles of different sizes in the upper 500 m of the Mauritanian upwelling zone in the North Atlantic Ocean (NAO)[55].** $O_2$ and $NO_3^-$ profile in the water column[55] (**a**), total amount of fixed $N_2$ per particle (**b**), fraction of particle volume where respiration is fueled by denitrification (**c**), and fraction of particle volume where respiration is fueled by $SO_4^{2-}$ reduction (**d**). Areas enclosed by white and magenta dashed lines represent the occurrence of $N_2$ fixation ($>10^{-3}$ µg N day$^{-1}$ particle$^{-1}$) and anoxia ($O_2 < 10^{-3}$ µmol $O_2$ L$^{-1}$), respectively, at the center of particles.

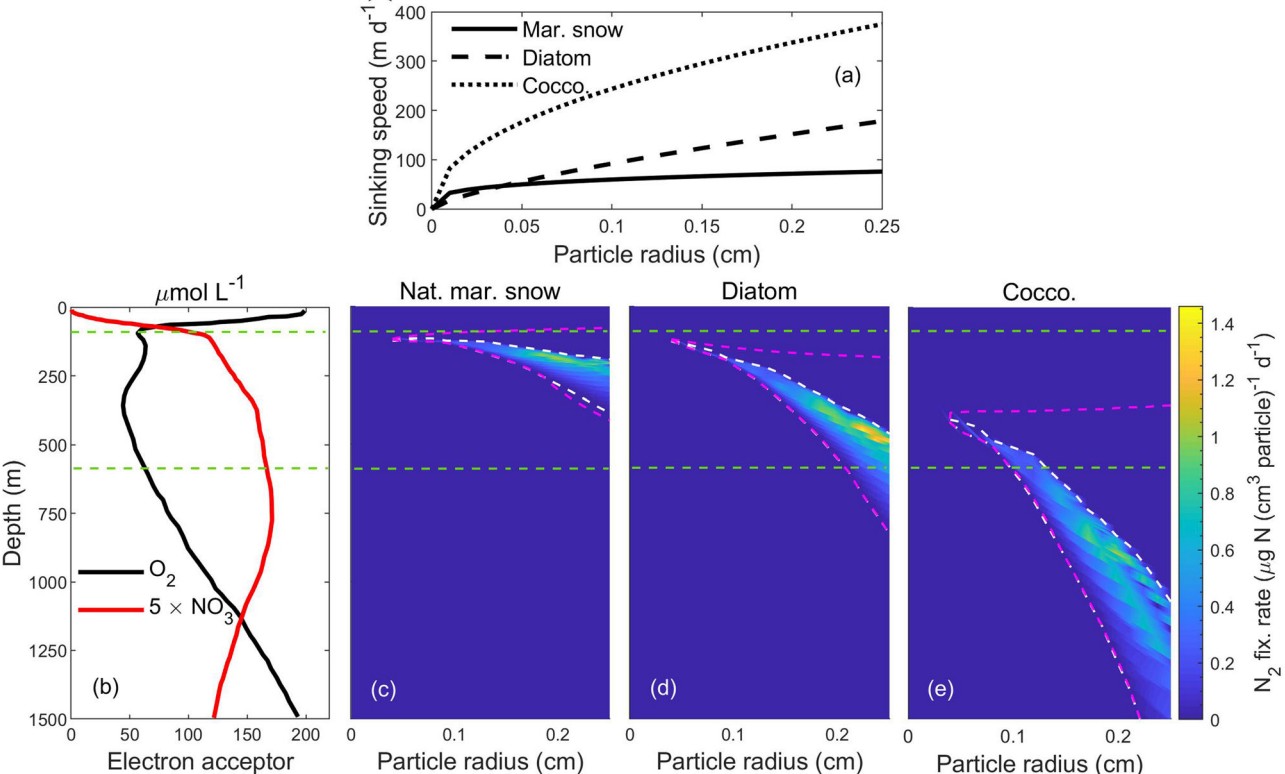

**Fig. 7 $N_2$ fixation rates in different types of sinking particles with different sinking speeds. a** Particle radius (cm) versus sinking speeds (m d$^{-1}$) of natural marine snow (continuous line) measured in situ off California ($v = 108.95 \times r_p^{0.26}$;[61]) and of laboratory-made diatom (dashed line) and coccolithophore (dotted line) aggregates measured in vitro ($v = 484.09 \times r_p^{0.72}$ and $v = 719.44 \times r_p^{0.47}$, respectively)[50]. **b** $O_2$ and $NO_3^-$ profiles of the Mauritanian upwelling zone in the North Atlantic Ocean (NAO)[55]. **c–e** $N_2$ fixation rates per unit volume of particles of different sizes and types. White and magenta dashed lines are same as in Fig. 6. The area enclosed within the horizontal green lines represents the low-$O_2$ zone.

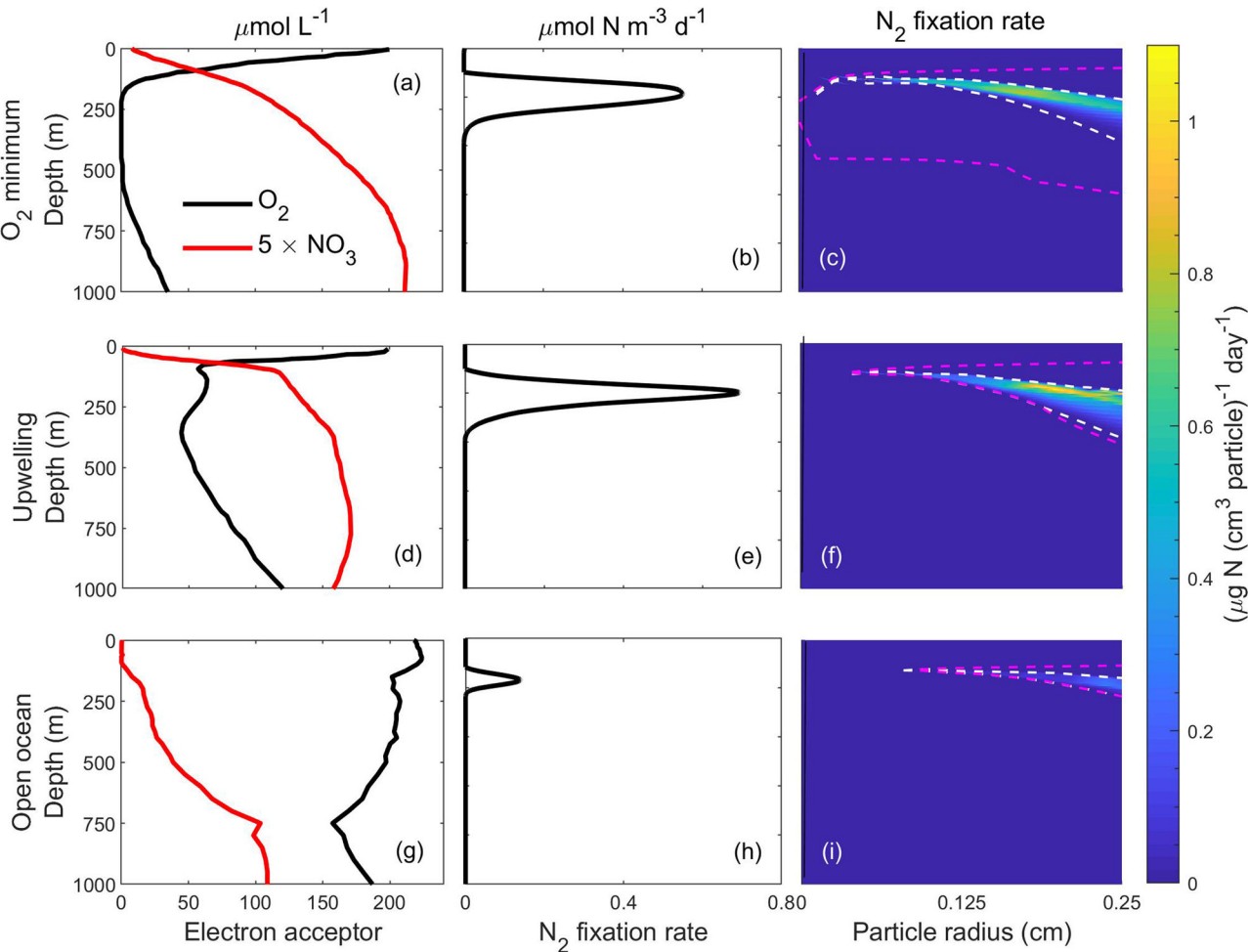

**Fig. 8 Comparison of predicted $N_2$ fixation rates in natural marine snow at three contrasting sites in terms of vertical distributions of $O_2$ and $NO_3^-$ in the ocean. a–c** $O_2$ minimum zone in the Eastern Tropical South Pacific (ETSP)[55], **d–f** the Mauritanian upwelling zone in the North Atlantic Ocean (NAO)[55], and **g–i** open ocean (OO; 30.5°N, 52.5°W)[62,63]. **a, d, g** $O_2$ and $NO_3^-$ concentrations in the upper 1000 m vertical water column. **b, e, h** $N_2$ fixation rates per unit volume of water. **c, f, i** $N_2$ fixation rates per unit volume of particle of different size classes.

the higher energy yield of $NO_3^-$ respiration relative to $SO_4^{2-}$ reduction, we speculate that higher concentrations of $NO_3^-$ during the time of $N_2$ fixation boosts $N_2$ fixation rate in diatom aggregates. High sinking speed, similar to that of coccolithophore aggregates, transports particles quickly to deep water with high $NO_3^-$ and $O_2$ concentrations (Fig. 7e). We expect that high $NO_3^-$ concentration favors relatively high $N_2$ fixation, but high $O_2$ concentration lowers the $N_2$ fixation rate in coccolithophore aggregates. Therefore, the interplay between particle sinking speed and vertical water column profiles of $O_2$ and $NO_3^-$ concentrations determines the $N_2$ fixation rate, supporting our hypothesis H5.

**$N_2$ fixation in contrasting oceanic environments.** $N_2$ fixation is dependent on the $O_2$ and $NO_3^-$ concentrations in the water column, but those are highly variable around the global ocean. We investigated $N_2$ fixation rates in three contrasting water columns: an $O_2$ minimum zone in the Eastern Tropical South Pacific (ETSP)[55], the Mauritanian upwelling zone in the NAO[55], and an open ocean site (OO; 30.5° N, 52.5° W)[62,63]. The ETSP has $O_2$ minimum zones with <5 μmol $O_2$ $L^{-1}$ at ~150 to 600 m depth (Fig. 8a), the NAO has reduced $O_2$ levels (~62.5 to 157 μmol $L^{-1}$) at ~100 to 600 m (Fig. 8d), whereas the open ocean site has high $O_2$ concentration throughout the water column (>157 μmol $L^{-1}$; Fig. 8g). $NO_3^-$ concentrations increase gradually up to 800 m depth, with decreasing levels from ETSP to NAO and to the open

ocean site. We consider particles with sinking speed similar to natural marine snow and examine $N_2$ fixation rates per unit volume of particle (Fig. 8c, f, i). To compare with existing measurements, we further calculate $N_2$ fixation rates per unit volume of water (Fig. 8b, e, h) and depth-integrated $N_2$ fixation rates by multiplying the number of particles with $N_2$ fixation per particle (Eq. 40)[34].

Maximum $N_2$ fixation rates per volume of particle and per volume of water lie within ranges 0.31–1.1 μg N (cm³ particle)$^{-1}$ d$^{-1}$ (Fig. 8c, f, i) and 0.14–0.7 μmol N m$^{-3}$ d$^{-1}$ (Fig. 8b, e, h). The highest rate of $N_2$ fixation is in the NAO followed by the ETSP. Interestingly, $N_2$ fixation is observed even at the high $O_2$ concentrations of the open ocean, although, the $N_2$ fixation rates and the depth window of $N_2$ fixation are smaller than for the other two scenarios. Since the abundance and size spectrum of particles vary with latitude and seasonally at high latitudes[64], we test the sensitivity of $N_2$ fixation rates by varying the parameter determining the abundance ($n_0$) and the proportion of large and small particles ($\xi$). We find that the $N_2$ fixation rate varies between 0.3 and 1.7 μmol N m$^{-3}$ d$^{-1}$ inside sinking particles (Supplementary Fig. S6). Our modeled $N_2$ fixation rates are comparable with bulk $N_2$ fixation rates measured in the aphotic ocean (0–0.89 μmol N m$^{-3}$ d$^{-1}$)[9,65] where active autotrophic cyanobacterial diazotrophs are not expected. Our calculated depth-integrated $N_2$ fixation rates lie within the range

7.1–65.1 µmol N m$^{-2}$ d$^{-1}$. Empirical evidence from regions, where N$_2$ fixation is dominated by heterotrophic bacteria, shows similar levels of depth-integrated N$_2$ fixation rates; e.g., 6.27–16.6 µmol N m$^{-2}$ d$^{-1}$ in the equatorial and southern Indian Ocean[66] and 12.4–190.9 µmol N m$^{-2}$ d$^{-1}$ in the South Pacific Gyre[67]. In comparison, depth-integrated N$_2$ fixation rates by cyanobacteria in most regions of the global upper ocean are in the order of 1–100 µmol N m$^{-2}$ d$^{-1}$ [68]. Hence, taken together, our modeled rates for heterotrophic bacteria on particles are consistent with empirical bulk rates from the deep sea and comparable to areal rates measured for cyanobacteria. This supports empirical studies suggesting that aphotic fixation can account for a significant or even predominant fraction of water column N$_2$ fixation[65,69], and substantiates the idea that aphotic N$_2$ fixation may be important to global nitrogen budget considerations[5].

**The effects of temperature variation**. The variation in water temperature has the potential to alter N$_2$ fixation rates through increases in metabolic rates and diffusive processes with temperature. Since sinking particles face a decreasing gradient in temperature as they descend through the water column, especially in the lower latitudes, we examined the effects of temperature on the amount of fixed N$_2$ at an open ocean site in the NAO (OO; 30.5° N, 52.5° W)[62,63]. A small increase in the amount of total N$_2$ fixation rate, from 7.1 µmol N m$^{-2}$ d$^{-1}$ to 8.4 µmol N m$^{-2}$ d$^{-1}$, and a slight downward shift in the positioning of N$_2$ fixation in the water column was observed (Supplementary Fig. S7). On one hand, elevated temperature increases N$_2$ fixation rate by stimulating metabolic activity. On the other hand, elevated temperature stimulates the influx of O$_2$ via diffusion, which hampers N$_2$ fixation. Therefore, the impact of temperature on metabolic rates and diffusion tends to counter one another on sinking particles. Hence, even at a site showing a large decrease in temperature between surface and depth, the effects of temperature on N$_2$ fixation are fairly small.

**Conclusions and broader implications**. Our model suggests that particle-associated heterotrophic N$_2$ fixation is viable and reasonable based on the known properties and physics of marine particles and reveals a significant contribution to the oceanic biological N$_2$ fixation. The likelihood and rate of N$_2$ fixation associated with any individual particle will depend upon numerous factors. These include the vertical profiles of O$_2$ and NO$_3^-$ through which the particle sinks, but in particular also the particle composition and bioavailability, including the initial polysaccharide and polypeptide concentrations, together with the size, sinking speed, and abundance of particles, as suggested by the model sensitivity analysis. We show how low-O$_2$ or anoxic zones generated inside sinking particles by microbial respiration provide conditions suitable for heterotrophic N$_2$ fixation, however, only in particles larger than about 0.03 cm in radius. Moreover, we show that these anoxic microenvironments can promote anaerobic respiratory processes that even extend into well-oxygenated deep waters. Interestingly, our simulations suggest that even in particles that favor N$_2$ fixation at a point in their descent, the window of time (depth) where diazotrophy occurs is likely to be short. However, despite the necessity of several coinciding environmental conditions for N$_2$ fixation in particles, the criteria are met in natural particles. Because of the huge number and heterogeneity among particles sinking in the ocean[43], it is highly likely that a large number of individual particles at any given time might meet these criteria and in doing so, confer a fitness advantage on a subset of bacteria that retain the ability to fix nitrogen. This may explain the ubiquity and persistence of the genetic signature for heterotrophic N$_2$ fixation

throughout the oceans[8]. The combined knowledge of the probability density of particle sizes, compositions, and sinking speeds is suggested to predict the average rates of N$_2$ fixation associated with particles.

Our model makes several interesting and potentially testable predictions: First, the C:N composition of particle "trails" will reflect the interior state and might be a way to probe the response of different particle types or conditions. Second, the model predicts a "preference" for SO$_4^{2-}$ electron acceptor over NO$_3^-$ in low-oxygen particles. Third, heterotrophic diazotrophs mostly use local and ephemeral oxygen conditions and get windows of opportunity for their N$_2$ fixation. Fourth, N$_2$ fixation can occur on large particles with high concentrations of polysaccharides and polypeptides in fully oxygenated marine waters, but also on older less substrate-rich particles if oxygen concentration in the surrounding water is low. These predictions could be promoted as perspectives for future experiments.

## Methods

### The cell model

*Growth rate of a cell*. The growth rate of a bacteria cell depends on the acquisition of C (from the particle) and N (from the particle and through N$_2$ fixation), as well as on metabolic expenses in terms of C.

*Uptake of C and N*. Bacteria get C from glucose and both C and N from amino acids. The total amount of C available for the cell from monomers is (units of C per time)

$$J_{DOC} = f_{G,C}J_G + f_{A,C}J_A, \qquad (8)$$

and the amount of N available from monomer is (N per time)

$$J_{DON} = f_{A,N}J_A, \qquad (9)$$

where $J_G$ and $J_A$ are uptake rates of glucose and amino acids, $f_{G,C}$ is the fraction of C in glucose, and $f_{A,C}$ and $f_{A,N}$ are fractions of C and N in amino acids.

The rate of obtaining N through N$_2$ fixation is:

$$J_{N_2}(\psi) = \psi M_{N_2}, \qquad (10)$$

where $\psi$ $(0<\psi<1)$ regulates N$_2$ fixation rate and fixation can happen at a maximum rate $M_{N_2}$. N$_2$ fixation is only limited by the maximum N$_2$ fixation rate as dissolved dinitrogen (N$_2$) gas in seawater is assumed to be unlimited[70].

The total uptake of C and N from different sources becomes

$$J_C = J_{DOC} \qquad (11)$$

$$J_N(\psi) = J_{DON} + J_{N_2}(\psi) \qquad (12)$$

*Costs*. Respiratory costs of cellular processes together with N$_2$ fixation and its associated O$_2$ removal cost depend on the cellular O$_2$ concentration. Two possible scenarios can be observed:

*Case 1: When O$_2$ concentration is sufficient to maintain aerobic respiration*

Respiratory costs for bacterial cellular maintenance can be divided into two parts: one dependent on limiting substrates and the other one is independent of substrate concentration[71]. Here we consider only the basal respiratory cost $R_Bx_B$, which is independent of the limiting substrates and is assumed as proportional to the mass of the cell $x_B$ (µg C). In order to solubilize particles, particle-attached bacteria produce ectoenzymes that cleave bonds to make molecules small enough to be transported across the bacterial cell membrane. Cleavage is represented by a biomass-specific ectoenzyme production cost $R_E$[72]. The metabolic costs associated with the uptake of hydrolysis products and intracellular processing are assumed to be proportional to the uptake ($J_i$): $R_GJ_G$ and $R_AJ_A$ where the $R_i$'s are costs per unit of resource uptake. In a similar way, the metabolic cost of N$_2$ fixation is assumed as proportional to the N$_2$ fixation rate: $R_{N_2}\rho_{CN,B}J_{N_2}$, where $\rho_{CN,B}$ is the bacterial C:N ratio. If we define all the above costs as direct costs, then the total direct respiratory cost becomes

$$R_D(\psi) = R_Bx_B + R_Ex_B + R_GJ_G + R_AJ_A + R_{N_2}\rho_{CN,B}J_{N_2}(\psi). \qquad (13)$$

Indirect costs related to N$_2$ fixation arises from the removal of O$_2$ from the cell and the production/replenishment of nitrogenase as the enzyme is damaged by O$_2$. The cell can remove O$_2$ either by increasing respiration[73] or by increasing the production of nitrogenase enzyme itself[74]. Here we consider only the process of O$_2$ removal by increasing respiration. To calculate this indirect cost, the concentration of O$_2$ present in the cell needs to be estimated.

Since the time scale of O$_2$ concentration inside a cell is short, we have assumed a pseudo steady state inside the cell; the O$_2$ diffusion rate inside a cell is always

balanced by the respiration rate[14], which can be expressed as

$$\rho_{CO} F_{O_2} = R_D(\psi). \tag{14}$$

Here $\rho_{CO}$ is the conversion factor of respiratory $O_2$ to C equivalents and $F_{O_2}$ is the actual $O_2$ diffusion rate into a cell from the particle and can be calculated as

$$F_{O_2} = 4\pi r_B K_{O_2}\left(X_{O_2} - X_{O_2,C}\right), \tag{15}$$

where $r_B$ is the cell radius, $X_{O_2}$ is the local $O_2$ concentration inside the particle, $XO_2,C$ is the cellular $O_2$ concentration, and $K_{O_2}$ is the effective diffusion coefficient of $O_2$ over cell membrane layers. The effective diffusion coefficient can be calculated according to Inomura et al.[14] in terms of diffusion coefficient inside particles ($\bar{D}_{O_2}$), the diffusivity of cell membrane layers relative to water ($\varepsilon_m$), the radius of cellular cytoplasm ($r_C$), and the thickness of cell membrane layers ($L_m$) as

$$K_{O_2} = \bar{D}_{O_2}\frac{\varepsilon_m(r_C + L_m)}{\varepsilon_m r_C + L_m}. \tag{16}$$

The apparent diffusivity inside particles ($\bar{D}_{O_2}$) is considered as a fraction $f_{O_2}$ of the diffusion coefficient in seawater ($D_{O_2}$)

$$\bar{D}_{O_2} = f_{O_2} D_{O_2}. \tag{17}$$

Combining (14) and (15) gives the cellular $O_2$ concentration $X_{O_2,C}$ as

$$X_{O_2,C} = \max\left[0, X_{O_2} - \frac{R_D(\psi)}{4\pi r_B K_{O_2}\rho_{CO}}\right]. \tag{18}$$

If there is excess $O_2$ present in the cell after respiration ($X_{O_2,C} > 0$), then the indirect cost of removing the excess $O_2$ to be able to perform $N_2$ fixation can be written as

$$R_{O_2}(\psi) = H(\psi)\rho_{CO} 4\pi r_B K_{O_2} XO_2, C, \tag{19}$$

where $H(\psi)$ is the Heaviside function:

$$H(\psi) = \begin{cases} 0, & \text{if } \psi = 0 \\ 1, & \text{if } \psi > 0 \end{cases}. \tag{20}$$

Therefore, the total aerobic respiratory cost becomes:

$$R_{tot,A}(\psi) = R_D(\psi) + R_{O_2}(\psi). \tag{21}$$

*Case 2: Anaerobic respiration*

When available $O_2$ is insufficient to maintain aerobic respiration ($R_{tot}(\psi) > \rho_{CO} F_{O_2,max}$), cells use $NO_3^-$ and $SO_4^{2-}$ for respiration. The potential $NO_3^-$ uptake, $J_{NO_3,pot}$, is

$$J_{NO_3,pot} = M_{NO_3}\frac{A_{NO_3}X_{NO_3}}{A_{NO_3}X_{NO_3} + M_{NO_3}}, \tag{22}$$

where $M_{NO_3}$ and $A_{NO_3}$ are maximum uptake rate and affinity for $NO_3^-$ uptake, respectively. However, the actual rate of $NO_3^-$ uptake, $J_{NO_3}$, is determined by cellular respiration and can be written as

$$J_{NO_3} = \min\left(J_{NO_3,pot}, \max\left(0, \frac{R_{tot,A}(\psi) - \rho_{CO}F_{O_2,max}}{\rho_{CNO_3}}\right)\right), \tag{23}$$

where $\rho_{CNO_3}$ is the conversion factor of respiratory $NO_3^-$ to C equivalents and the maximum $O_2$ diffusion rate into a cell $F_{O_2,max}$ can be obtained by making cellular $O_2$ concentration $X_{O_2,c}$ zero in (15) as

$$F_{O_2,max} = 4\pi r_B K_{O_2} X_{O_2}, \tag{24}$$

Further, in the absence of sufficient $NO_3^-$, the cell uses $SO_4^{2-}$ as an electron acceptor for respiration. Since the average concentration of $SO_4^{2-}$ in seawater is 29 mmol $L^{-1}$ [75], $SO_4^{2-}$ is a nonlimiting nutrient for cell growth and the potential uptake rate of $SO_4^{2-}$ is mainly governed by the maximum uptake rate as

$$J_{SO_4,pot} = M_{SO_4}, \tag{25}$$

where $M_{SO_4}$ is the maximum uptake rate for $SO_4^{2-}$ uptake. The actual rate of $SO_4^{2-}$ uptake, $J_{SO_4}$, can be written as

$$J_{SO_4} = \min\left(J_{SO_4,pot}, \max\left(0, \frac{R_{tot,A}(\psi) - \rho_{CO}F_{O_2,max} - \rho_{CNO_3}F_{NO_3,pot}}{\rho_{CSO_4}}\right)\right), \tag{26}$$

where $\rho_{CSO_4}$ is the conversion factor of respiratory $SO_4^{2-}$ to C equivalents.

According to formulations (23) and (26), $NO_3^-$ and $SO_4^{2-}$ uptake occurs only when the diffusive flux of $O_2$, and both $O_2$ and $NO_3^-$ are insufficient to maintain respiration. Moreover, the uptake rates of $NO_3^-$ and $SO_4^{2-}$ are regulated according to the cells' requirements.

Uptakes of $NO_3^-$ and $SO_4^{2-}$ incur extra metabolic costs $R_{NO_3}\rho_{CNO_3}J_{NO_3}$ and $R_{SO_4}\rho_{CSO_4}J_{SO_4}$, where $R_{NO_3}$ and $R_{SO_4}$ are costs per unit of $NO_3^-$ and $SO_4^{2-}$ uptake.

The total respiratory cost can be written as

$$R_{tot}(\psi) = R_{tot,A}(\psi) + R_{NO_3}\rho_{CNO_3}J_{NO_3} + R_{SO_4}\rho_{CSO_4}J_{SO_4}. \tag{27}$$

*Synthesis and growth rate.* The assimilated C and N are combined to synthesize new structure. The synthesis rate is constrained by the limiting resource (Liebig's law of the minimum) and by available electron acceptors such that the total flux of C available for growth $J_{tot}$ (µg C $d^{-1}$) is:

$$J_{tot}(\psi) = \min\left[J_C - R_{tot}(\psi), \rho_{CN,B}J_N(\psi), \rho_{CO}F_{O_2} + \rho_{CNO_3}J_{NO_3} + \rho_{CSO_4}J_{SO_4}\right]. \tag{28}$$

Here, the total available C for growth is $J_C - R_{tot}(\psi)$, the C required to synthesize biomass from N source is $\rho_{CN,B}JN$, and the C equivalent inflow rate of electron acceptors to the cell is $\rho_{CO}F_{O_2} + \rho_{CNO_3}J_{NO_3} + \rho_{CSO_4}J_{SO_4}$. We assume that excess C or N is released from the cell instantaneously.

Synthesis is not explicitly limited by a maximum synthesis capacity; synthesis is constrained by the C and N uptake in the functional responses (Eqs. 34 and 35). The division rate $\mu$ of the cell ($d^{-1}$) is the total flux of C available for growth divided by the C mass of the cell ($x_B$):

$$\mu(\psi) = J_{tot}(\psi)/x_B. \tag{29}$$

The resulting division rate, $\mu$, is a measure of the bacterial fitness and we assume that the cell regulates its $N_2$ fixation rate depending on the environmental conditions to gain additional N while maximizing its growth rate. The optimal value of the parameter regulating $N_2$ fixation $\psi$ ($0 \leq \psi \leq 1$) then becomes:

$$\psi^* = \arg\max_\psi\{\mu(\psi)\}, \tag{30}$$

and the corresponding optimal division rate becomes

$$\mu^* = \mu\left(\psi^*\right). \tag{31}$$

**The particle model**. We consider a sinking particle of radius $r_P$ (cm) and volume $V_P$ ($cm^3$) (Supplementary Fig. S1). The particle contains facultative nitrogen-fixing bacterial population $B(r)$ (cells $L^{-1}$), polysaccharides $C_P(r)$ (µg G $L^{-1}$), and polypeptides $P_P(r)$ (µg A $L^{-1}$) at a radial distance $r$ (cm) from the center of the particle, where G and A stand for glucose and amino acids. We assume that only fractions $f_C$ and $f_P$ of these polymers are labile ($C_L(r) = f_C C_P(r)$, $P_L(r) = f_P P_P(r)$), i.e., accessible by bacteria. Bacterial enzymatic hydrolysis converts the labile polysaccharides and polypeptides into monosaccharides (glucose) ($G$ µg G $L^{-1}$) and amino acids ($A$ µg A $L^{-1}$) that are efficiently taken up by bacteria. Moreover, the particle contains $O_2$, $NO_3^-$, and $SO_4^{2-}$ with concentrations $X_{O_2}(r)$ (µmol $O_2$ $L^{-1}$), $X_{NO_3}(r)$ (µmol $NO_3$ $L^{-1}$), and $X_{SO_4}(r)$ (µmol $SO_4$ $L^{-1}$). Glucose and amino acids diffuse out of the particle whereas $O_2$ and $NO_3^-$ diffuse into the particle from the surrounding environment. Due to the high concentration of $SO_4^{2-}$ in ocean waters, we assume that $SO_4^{2-}$ is not diffusion limited inside particles, its uptake is limited by the maximum uptake capacity due to physical constraint. The interactions between particle, cells, and the surrounding environment are explained in Supplementary Fig. S1 and equations are provided in Table 1 of the main text.

We assume that labile polysaccharide ($C_L$) and polypeptide ($P_L$) are hydrolyzed into glucose and amino acids at rates $J_C$ and $J_P$ with the following functional form

$$J_C = h_C \frac{A_C C_L}{h_C + A_C C_L} \tag{32}$$

$$J_P = h_P \frac{A_P P_L}{h_P + A_P P_L} \tag{33}$$

where $h_C$ and $h_P$ are maximum hydrolysis rates of the carbohydrate and peptide pool, and $A_C$ and $A_P$ are respective affinities. $J_G$ and $J_A$ represent uptake of glucose and amino acids:

$$J_G = M_G \frac{A_G G}{A_G G + M_G} \tag{34}$$

$$J_A = M_A \frac{A_A A}{A_A A + M_A} \tag{35}$$

where $M_G$ and $M_A$ are maximum uptake rates of glucose and amino acids, whereas $A_G$ and $A_A$ are corresponding affinities. Hydrolyzed monomers diffuse out of the particle at a rate $D_M$.

$\mu^*$ is the optimal division rate of cells (Eq. 31) and $m_B$ represents the mortality rate (including predation) of bacteria. $F_{O_2}$ and $J_{NO_3}$ represent the diffusive flux of $O_2$ and the consumption rate of $NO_3^-$, respectively, through the bacterial cell membrane. $\bar{D}_{O_2}$ and $\bar{D}_{NO_3}$ are diffusion coefficients of $O_2$ and $NO_3^-$ inside the particle.

At the center of the particle ($r = 0$) the gradient of all quantities vanishes:

$$\left.\frac{\partial G}{\partial r}\right|_{r=0} = \left.\frac{\partial A}{\partial r}\right|_{r=0} = \left.\frac{\partial X_{O_2}}{\partial r}\right|_{r=0} = \left.\frac{\partial X_{NO_3}}{\partial r}\right|_{r=0} = 0 \tag{36}$$

At the surface of the particle ($r = r_P$) concentrations are determined by the surrounding environment:

$$G|_{r=r_P} = G_\infty, A|_{r=r_P} = A_\infty, X_{O_2}\big|_{r=r_P} = X_{O_2,\infty}, X_{NO_3}\big|_{r=r_P} = X_{NO_3,\infty} \quad (37)$$

where $G_\infty$, $A_\infty$, $X_{O_2,\infty}$ and $X_{NO_3,\infty}$ are concentrations of glucose, amino acids, $O_2$, and $NO_3^-$ in the environment.

**Calculation of total N$_2$ fixation rate.** The total amount of fixed $N_2$ in a specific size class of particle, $N_{fix,P}$ (µg N particle$^{-1}$), is calculated as

$$N_{fix,P} = \int\int 4\pi r_B^2 B J_{N_2} \, dr_P dz, \quad (38)$$

where $r_P$ (cm) is the particle radius and z (m) represents the water column depth.

$N_2$ fixation rate per unit volume of water, $N_{fix,V}(t)$ (µmol N m$^{-3}$ d$^{-1}$), is calculated as

$$N_{fix,V} = \int\int 4\pi r_B^2 \rho B J_{N_2} n(x) \, dr_P dx, \quad (39)$$

Here $x$ (cm) represents the size range (radius) of particles, $\rho$ is the fraction of diazotrophs of the total heterotrophic bacteria, and $n(x)$ (number of particles per unit volume of water per size increment) is the size spectrum of particles that is most commonly approximated by a power law distribution of the form

$$n(x) = n_0(2x)^\xi \quad (40)$$

where $n_0$ is a constant that controls total particle abundance and the slope $\xi$ represents the relative concentration of small to large particles: the steeper the slope, the greater the proportion of smaller particles and the flatter the slope, and the greater the proportion of larger particles[34].

Depth-integrated $N_2$ fixation rate, $N_{fix,D}$ (µmol N m$^{-2}$ d$^{-1}$), can be obtained by

$$N_{fix,D}(t) = \int N_{fix,V} dz. \quad (41)$$

**Assumptions and simplification in the modeling approach.** According to our current model formulation, the particle size remains constant while sinking. However, in nature, particle size is dynamic due to processes like bacterial remineralization, aggregation, and disaggregation. We neglect these complications to keep the model simple and to focus on revealing the coupling between particle-associated environmental conditions and $N_2$ fixation by heterotrophic bacteria. These factors can, however, possibly be incorporated by using in situ data or by using the relationship between carbon content and the diameter of particles[48] and including terms for aggregation and disaggregation[55].

Our model represents a population of facultative heterotrophic diazotrophs that grow at a rate similar to other heterotrophic bacteria but the whole community initiates $N_2$ fixation when conditions become suitable. However, under natural conditions, diazotrophs may only constitute a fraction of the bacterial community, and their proliferation may be gradual[21], presumably affected by multiple factors. In such case, our approach will overestimate diazotroph cell concentration and consequently the $N_2$ fixation rate.

For simplicity, our approach includes only aerobic respiration, $NO_3^-$ and $SO_4^{2-}$ respiration, although many additional aerobic and anaerobic processes likely occur on particles (e.g Klawonn et al.[19]). To our knowledge, a complete picture of such processes, their interactions and effects on particle biochemistry is unavailable. For example, we have assumed that when $O_2$ and $NO_3^-$ are insufficient to maintain respiration, heterotrophic bacteria start reducing $SO_4^{2-}$. However, $SO_4^{2-}$ reduction has been detected only with a significant lag after the occurrence of anaerobic conditions, suggesting it as a slow adapted process[76], whereas we assume it to be instantaneous. On the other hand, the lag may not be real but due to a so called cryptic sulfur cycle, where $SO_4^{2-}$ reduction is accompanied by concurrent sulfide oxidation effectively masking sulfide production[77]. Hopefully, future insights into interactions between diverse aerobic and anaerobic microbial processes can refine our modelling approach and fine-tune predictions of biochemistry in marine particles.

**Procedure of numerically obtaining optimal N$_2$ fixation rate.** To avoid making the optimization in Eq. (30) at every time step during the simulation, a lookup table of $\mu^*$ (Eq. 31) over realistic ranges of the four resources (glucose, amino acids, $O_2$, and $NO_3^-$) and the parameter determining $N_2$ fixation rate ($\psi$) was created at the beginning of the simulation.

**The effects of temperature on N$_2$ fixation rate.** To examine the role of temperature variation on $N_2$ fixation rate in sinking particles, we consider hydrolysis of polysaccharide and polypeptide, uptake of glucose and amino acids, uptake of $NO_3^-$, respiration, and diffusion dependent on temperature. Apart from diffusion, all other processes are multiplied by a factor $Q_{10}$ that represents the factorial increase in rates with 10$^0$C temperature increase. The rate $R$ at a given temperature

$T$ is then

$$R = R_{ref}Q_{10}^{(T-T_{ref})/10}. \quad (42)$$

Here the reference rate $R_{ref}$ is defined as the rate at the reference temperature $T_{ref}$. We set the reference temperature $T_{ref}$ at room temperature of 20 °C. The effect of temperature on the diffusion coefficient $D$ for glucose, amino acids, $O_2$, and $NO_3^-$ is described by Walden's rule:

$$D = D_{ref}\eta_{ref}T/(\eta T_{ref}) \quad (43)$$

where $\eta$ is the viscosity of water at the given temperature $T$, and $D_{ref}$ and $\eta_{ref}$ are diffusion coefficient and viscosity at $T_{ref}$.

$Q_{10}$ values for different enzyme classes responsible for hydrolysis ($Q_{10,h}$) lie within the range 1.1–2.9[78]. Here, we have chosen $Q_{10,h} = 2$ for hydrolysis from the middle of the prescribed range. The $Q_{10}$ values for uptake affinities ($Q_{10,A}$) are taken as 1.5[79]. $Q_{10,R} = 2$ is chosen for all parameters related to respiration ($R_B$, $R_E$, $R_G$, $R_A$, $R_{N_2}$, $R_{NO_3}$, $R_{SO_4}$)[80]. $R_{ref}$ and $D_{ref}$ are the values of $R$'s and $D$'s provided in Table S1. The reference viscosity ($\eta_{ref}$) and viscosities ($\eta$) at different temperatures are taken from Jumars et al.[80].

## Data availability

The authors declare that the sources of all data supporting the findings of this study are available within the article.

## Code availability

Matlab codes for Figs. 2 and 3 are available at https://doi.org/10.17894/ucph.da390b38-13d1-40a3-a287-9f8742494d65 (https://erda.ku.dk/archives/1e3ce4f581c38fb00bb4311126a9b513/published-archive.html).

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

## Acknowledgements

S.C. and L.R. were supported by grant 6108-00013 from the Danish Council for Independent Research to L.R. The Centre for Ocean Life was supported by the Villum Foundation. K.I. was supported by the Simons Foundation (Simons Postdoctoral Fellowship in Marine Microbial Ecology, Award 544338). M.J.F. was supported by the Simons Collaboration on Computational Biogeochemical Modeling of Marine Ecosystems (CBIOMES; Simons Foundation grant no. 549931). We wish to thank Helle Ploug for her helpful comments on an earlier version of the manuscript.

## Author contributions

S.C., L.R., K.H.A., and A.W.V. developed the model. S.C. performed all numerical simulations and analyses with input from K.H.A., A.W.V., and K.I. S.C. and L.R. wrote the manuscript. S.C., K.H.A., A.W.V., K.I., M.J.F., and L.R. provided critical feedback and helped shape the manuscript.

## Competing interests

The authors declare no competing interests.
