## [Peer Review File · Nature Communications]

REVIEWER COMMENTS

Reviewer #1 (Remarks to the Author):

Chakraborty et al. present an analytical model of heterotrophic-based N₂ fixation associated with marine particles building on some of the models that co-author Inomura has previously presented.

The depth and the breadth of the model considerations are very impressive. The model is developed in two stages: a "Cell" model working at the metabolic level considering uptake of nutritional elements and electron acceptors, respiration and anabolic pathways and growth. The Cell model is embedded in a larger "Particle" model."

The models are run to tease apart the interacting influences of particle size, O₂ concentration, organic compounds (polysaccharides and polypeptides).

Several hypotheses are articulated and modeling runs are undertaken to demonstrate the minimum size of a particle which will support N₂ fixation, the mix of organics which provide an optimal conditions for development of a zone of low O₂ which will support N₂ fixation including the predominant metabolisms which provide the energy for this process within the particles – which often turns out to be sulfate reduction (rather than nitrate reduction/ denitrification).

The authors go on to examine particle associated N₂ fixation in a specific few oceanographic settings and using a set of three contrasting particle types of different settling speeds (marine snow, diatom aggregates and coccothithophore aggregates), and specifically how N₂ fix in these particles might evolve over time and depth.

I suppose the one minor issue I have is regarding the H1 hypothesis and its further discussion in the manuscript (e.g. lines 394-395). While it may be correct, I don't see anything in the modeling output- or empirical evidence referenced- to substantiate this piece. Yes, there's a mention of excess N on lines 189-191. Perhaps this aspect could be better supported.

The role of heterotrophic N₂ fixation in the sea, how and where it occurs and its quantitative significance is highly topical. The present manuscript brings together many of the pressing current questions with clear directions for future research to be tackled.

Overall, the manuscript is very well written and logically presented.

Specific Comments:

Line 156. Regarding the 0.125 cm radius particle discussed here and in the Figures 2 & 3 and used as a benchmark. There appears to be a minor inconsistency. In the Figure 2 legend, a radius of 0.25cm is given. However, the Y axis cm in Fig. 2 goes from 0 to 0.125cm. I assume the authors mean a diameter of 0.25cm here and as implied in Fig. 3.

Line 168. Perhaps reword? "... seldom occurs within sinking particles".

Line 173-174. The point is made earlier that the energetic yield from NO₃⁻ respiration is only incrementally smaller than that for O₂.

p. 15, bottom lines 330-335- seems to run counter to earlier conclusions regarding NO₃ respiration. That high [NO₃⁻] favor increased N₂ fixation is also somewhat counterintuitive although there's now sufficient field observations to uphold that diazotrophs and N₂ fixation can occurring in NO₃ rich environments.

Line 349. "Fig." (caps)

Douglas G Capone

Reviewer #2 (Remarks to the Author):

This manuscript presents an interesting model about something that has been very difficult to measure in the field. It demonstrates that N-fixation could occur in particles larger than ~500 micrometer in diameter. The model shows a mechanism by which SO₄ rather than NO₃ is used relatively quickly which is intriguing.

The model appears to be well assembled, and seems to conceptually follow from an earlier paper by Inomura et al. 2017. Unfortunately, I believe that also some of the parameter values have stuck from that paper. And those may be not realistic for water column particles (see below). Temperature has a strong forcing effect on metabolic processes. I have not seen any reference to the typically changing temperatures with depth, which can be very simply modeled by applying, for instance, the water column characteristic of the tropical ocean with a permanent thermocline. Was changing temperature considered?

Oxygen diffusion rates are critical in whether or not the particles are becoming sufficiently anoxic to allow for SO₄ respiration. The diffusion coefficient is stated as the fraction of the diffusion coefficient is seawater but not clear from the paper what it is. Since it such a critical aspect, it needs to be stated and discussed in more detail.

Using diffusion values suitable for alginate pellets or biofilms is likely overestimating the barrier that the particle matrix represents.

The typical marine snow aggregates are likely not getting anoxic because of the lack of diffusion barriers but instead tightly packed fecal pellets are more likely to be sites of anaerobic processes. On the other hand, fecal pellets at 500 um and larger sink up to kilometers per day and may disappear from the modeled depth range (especially from the mesopelagic) quickly, and before N-fixation occurs.

While fecal pellets are the most likely locations of anoxia inside particles, they represent only a very small fraction of the total volume of particles found in the deep sea (especially below 1000 m), which again is a problem when considering the total amount of N-fixation that is possible.

While the diffusion limitation of oxygen is critical to maintaining an oxygen gradient in the particle, the authors then assume no diffusion limitation for SO₄ (line 532) which in my opinion does not make sense (if I understood correctly).

I am always wondering about the design of the model when I see abrupt changes in the modeling of natural processes. The sudden total loss of glucose and amino acids with a simultaneous sudden onset of N-fixation makes me wonder whether some assumptions are oversimplified. Sharp transitions are very rare in natural systems (with exceptions of course, e.g., quorum sensing).

Growth rates and changes of bacterial numbers are simply impossible (3.6 d⁻¹?). If the model needs growth rates that high to work, there is something wrong structurally. The change of bacterial numbers is much too fast for the decrease in bacterial numbers too (especially without predation). But are these abrupt changes in bacterial numbers even necessary for the model to work?

The narrative lacks a simple recap of the model, essentially a short stepwise explanation how the various factors lead to the outcome (perhaps a flow chart?). Beyond the mathematical outcome, does it make sense? The model has to work conceptually, therefore the authors should briefly report the causal sequence of events.

It would also be very useful for the reader to run a sensitivity analysis that shows which components most strongly affect the main outcomes (i.e., occurrence of nitrogen fixation inside the particle). This could be demonstrated by showing a matrix/table or heat map.

Line 168 Replace seldom with rare

Reviewer #3 (Remarks to the Author):

This manuscript presents a mathematical model of nitrogen fixation as an optimal growth strategy utilized by heterotrophic bacteria in sinking marine particles. The model is well constructed and appropriate for the questions addressed. The analyses are insightful and clearly presented. The paper is well written and addresses important questions in microbial ecology and global biogeochemistry. The authors have done fine work and I think the manuscript can be published as is.

28 March 2021

Dear Dr. Frischkorn,

Thank you for giving us the opportunity to respond to the comments by the reviewers. We also thank Prof. Capone and one anonymous reviewer for his/her positive and constructive comments and suggestions, and one reviewer for suggesting to accept the paper in its current form. We have taken all comments into account in the revision; replies to each of the queries are provided below in blue. A few overarching points are worth highlighting here:

- 1) Reviewer 2 raised concerns about the bacterial growth rates on particles and suggested that the model output is not realistic. However, there are a number of studies that show significantly higher growth rates than the reviewer implies, which are comparable to our model output. In the new manuscript, we cite some of these papers in the main manuscript. Our detailed response can be found below.
- 2) We have followed most of the suggestions by the reviewers. Specifically, we have carried out a sensitivity analysis, incorporated temperature to examine its effect on N₂ fixation, and added a conceptual flow chart of how different factors are responsible for different events inside particles. The main conclusions drawn in the original manuscript are unchanged, but the additional analyses have substantially improved the manuscript.

We hope you will find the revisions satisfactory but naturally remain open to any further suggestions.

Subhendu Chakraborty and Lasse Riemann

REVIEWER COMMENTS

Reviewer #1:

Chakraborty et al. present an analytical model of heterotrophic-based N₂ fixation associated with marine particles building on some of the models that co-author Inomura has previously presented. The depth and the breadth of the model considerations are very impressive.

Thank you for such an encouraging comment. Below, we carefully followed the suggestions, which we believe improved our manuscript, and we hope that the new version will be found satisfactory.

I suppose the one minor issue I have is regarding the H1 hypothesis and it's further discussion in the manuscript (e.g. lines 394-395). While it may be correct, I don't see anything in the modeling output– or empirical evidence referenced- to substantiate this piece. Yes, there's a mention of excess N on lines 189-191. Perhaps this aspect could be better supported.

We agree with the reviewer and have now provided detail to substantiate this issue, as requested. Specifically, we have added the following paragraph (lines 232-237):

“The identification of different phases yields insights relevant for several of our hypotheses. As the particle sinks, it will be tailed by an organic matter solute trail. Its composition depends on the internal state of the particle. During the ‘C limited phase’ and ‘High respiration phase’, the expected trail consists of both C and N, whereas during the ‘N₂ fixation phase’, the trail contains only C. The amount of C and N in the trail can be estimated from the outgoing flux of the organic matter from the particle, supporting our hypothesis H1”.

The role of heterotrophic N₂ fixation in the sea, how and where it occurs and its quantitative significance is highly topical. The present manuscript brings together many of the pressing current questions with clear directions for future research to be tackled. Overall, the manuscript is very well written and logically presented.

Thank you for acknowledging the significance of the paper.

Specific Comments:

Line 156. Regarding the 0.125 cm radius particle discussed here and in the Figures 2 & 3 and used as a benchmark. There appears to be a minor inconsistency. In the Figure 2 legend, a radius of 0.25cm is given. However, the Y axis cm in Fig. 2 goes from 0 to 0.125cm. I assume the authors mean a diameter of 0.25cm here and as implied in Fig. 3.

Revised as suggested.

Line 168. Perhaps reword? "... seldom occurs within sinking particles".

The sentence is reformulated (lines 171-172): "The occurrence of such a high growth rate^{40,41} and bacterial abundance^{25,42,43} is not rare inside natural sinking particles."

Line 173-174. The point is made earlier that the energetic yield from NO₃⁻ respiration is only incrementally smaller than that for O₂.

Yes, we agree. But it is indeed lower as pointed out in lines 77-80 and leads to an observed reduction in growth rate.

p. 15, bottom lines 330-335- seems to run counter to earlier conclusions regarding NO₃ respiration. That high [NO₃⁻] favor increased N₂ fixation is also somewhat counterintuitive although there's now sufficient field observations to uphold that diazotrophs and N₂ fixation can occurring in NO₃ rich environments.

We agree that at first glance this seems counterintuitive. To clarify this we already described how NO₃⁻ respiration happens only in a limited volume fraction of particles (lines 327-336). We expect that, in the presence of relatively higher concentrations of NO₃⁻, this volume fraction where NO₃⁻ respiration occurs will also increase and together with a higher energy yield of NO₃⁻, total N₂ fixation will be elevated. In the revised text we have done our best to describe this scenario in a clear manner (lines 361-364):

"Since the presence of higher concentrations of NO₃⁻ can stimulate NO₃⁻ respiration in a relatively larger volume fraction of particles, and due to the higher energy yield of NO₃⁻ respiration relative to SO₄²⁻ reduction, we speculate that higher concentrations of NO₃⁻ during the time of N₂ fixation boosts N₂ fixation rate in diatom aggregates."

Line 349. "Fig." (caps)

Revised as suggested.

Reviewer #2:

This manuscript presents an interesting model about something that has been very difficult to measure in the field.

Thank you for this encouraging comment. We have carefully considered each comment and provide the response below. Following the comments, we have revised the manuscript in the following manner:

1. We have tested the effect of temperature.
2. We have provided a flowchart showing how different factors are responsible for different events inside particles.
3. We have conducted a sensitivity analysis.

In addition, we have carefully revised the manuscript following each comment of the reviewer. We thank the reviewer for the many constructive suggestions, which we think have improved the manuscript significantly. We hope the revised version will be found satisfactory. Please, note that references mentioned in our responses are provided in full at the very end of the responses to Reviewer 2.

The model appears to be well assembled, and seems to conceptually follow from an earlier paper by Inomura et al. 2017. Unfortunately, I believe that also some of the parameter values have stuck from that paper. And those may be not realistic for water column particles (see below).

While our paper has some conceptual similarity to the earlier work of Inomura and coworkers on the soil bacterium *Azotobacter*, only one parameter has been carried over. This is the assumption of “diffusivity of cell membrane layers relative to water (ϵ_m)”. To the best of our knowledge, and as also stated by Inomura et al. (2018), this parameter is not available for any aquatic heterotrophic N_2 fixers. So the value for *Azotobacter*, which is a heterotrophic N_2 fixer, seems like the best available choice to start with.

In our revised manuscript version, we have performed and provided results of a sensitivity analysis of this parameter (together with other parameters; please see the reply for the specific comment on sensitivity analysis) to check how this parameter affects N_2 fixation rate per particle (Supplementary Fig. S2). We did not find significant effect of ϵ_m on the sensitivity of N_2 fixation rate.

Temperature has a strong forcing effect on metabolic processes. I have not seen any reference to the typically changing temperatures with depth, which can be very simply modeled by applying, for instance, the water column characteristic of the tropical ocean with a permanent thermocline. Was changing temperature considered?

Thank you for this relevant and constructive comment. Accordingly, we have now examined the effects of temperature variation on N_2 fixation through a vertical water column at an open ocean

site. We now explain the incorporation of temperature (methods section; lines 651-669), provide a figure in the supplementary material (Fig. S7), and discuss the outcome in the main text (lines 406-418):

“The variation in water temperature has the potential to alter N₂ fixation rates through increases in metabolic rates and diffusive processes with temperature. Since sinking particles face a decreasing gradient in temperature as they descend through the water column, especially in the lower latitudes, we examined the effects of temperature on the amount of fixed N₂ at an open ocean site in the North Atlantic Ocean (30.5°N, 52.5°W)^{63,64}. A small increase in the amount of total N₂ fixation rate, from 7.1 μmol N m⁻² d⁻¹ to 8.4 μmol N m⁻² d⁻¹, and a slight downward shift in the positioning of N₂ fixation in the water column was observed (Supplementary Fig. S7). On one hand, elevated temperature increases N₂ fixation rate by stimulating metabolic activity. On the other hand, elevated temperature stimulates the influx of O₂ via diffusion, which hampers N₂ fixation. Therefore, the impact of temperature on metabolic rates and diffusion tends to counter one another on sinking particles. Hence, even at a site showing a large decrease in temperature between surface and depth, the effects of temperature on N₂ fixation are fairly small.”

Oxygen diffusion rates are critical in whether or not the particles are becoming sufficiently anoxic to allow for SO₄ respiration. The diffusion coefficient is stated as the fraction of the diffusion coefficient is seawater but not clear from the paper what it is. Since it such a critical aspect, it needs to be stated and discussed in more detail.

Using diffusion values suitable for alginate pellets or biofilms is likely overestimating the barrier that the particle matrix represents.

We apologize for the lack of clarity. We have chosen the diffusion of gases inside particles as 0.95 times that of water. This is an average value originating from direct oxygen diffusion measurements carried out on marine diatom particles by Ploug and Passow (2007). To clarify this, we have added the following information to the section in the ‘Model Parameterization’ part (Supplementary materials, lines 109-115):

“Direct measurement of apparent diffusivity of O₂ within diatom aggregates using a diffusivity microsensor shows values of 0.90 to 0.95 times the free diffusion coefficient in seawater³⁰. In the present study, the diffusion coefficients of O₂ and NO₃⁻ inside particles (\bar{D}_{O_2} , \bar{D}_{NO_3}) are assumed 0.95 times that of the free diffusion coefficient in seawater (D_{O_2} , D_{NO_3}). Since, compact particles, like fecal pellets, can have less diffusivity inside particles, we performed a sensitivity analysis to examine how changes in apparent diffusivity (f_{O_2}) affect N₂ fixation in particles. This suggests that N₂ fixation rate will increase in compact particles with lower apparent diffusivity (Fig. S2).”

The typical marine snow aggregates are likely not getting anoxic because of the lack of diffusion barriers but instead tightly packed fecal pellets are more likely to be sites of anaerobic processes. On the other hand, fecal pellets at 500 μm and larger sink up to kilometers per day and may

disappear from the modeled depth range (especially from the mesopelagic) quickly, and before N-fixation occurs.

While fecal pellets are the most likely locations of anoxia inside particles, they represent only a very small fraction of the total volume of particles found in the deep sea (especially below 1000 m), which again is a problem when considering the total amount of N-fixation that is possible.

Thank you for the careful comment. One of the main goals of our work was to investigate N₂ fixation rates in different types of particles (represented by different polysaccharide and polypeptide concentrations) with different sizes at different O₂ concentrations. Our model suggests that under relatively high O₂ concentrations, N₂ fixation is possible only in large particles with high polysaccharide and polypeptide concentrations (Fig. 5a), similar to freshly formed dense diatom aggregates. Moreover, N₂ fixation can occur in relatively smaller and compact particles similar to fecal pellets but only under very low environmental O₂ concentrations (Fig. 5c). We have also shown that particles like natural marine snow start fixing N₂ around a depth of ~ 200 m. Hence, whereas we agree with the reviewer that fecal pellets presumably could be loci suitable for N₂ fixation due to their high density, the model suggests that other particles also provide suitable environment for N₂ fixation in a reasonable depth range.

While the diffusion limitation of oxygen is critical to maintaining an oxygen gradient in the particle, the authors then assume no diffusion limitation for SO₄ (line 532) which in my opinion does not make sense (if I understood correctly).

We appreciate the reviewer's comment. However, SO₄²⁻, like N₂, is plentiful in seawater, and is likely not limited, justifying our assumption. To clarify this point, we have revised the manuscript (lines 146-151):

“SO₄²⁻ is the second most abundant anion in seawater with an estimated concentration of 28 mM³⁸ (roughly three orders of magnitude higher concentration than NO₃⁻) whereas N₂ is also plentiful with an average concentration of 0.4 mM in seawater³⁸ (more than two orders of magnitude higher concentration than NO₃⁻). Due to these high concentrations of SO₄²⁻ and N₂ inside particles, the uptake is assumed to be limited by the cellular maximum uptake capacities and not by the rate of diffusion towards the cell.”

I am always wondering about the design of the model when I see abrupt changes in the modeling of natural processes. The sudden total loss of glucose and amino acids with a simultaneous sudden onset of N-fixation makes me wonder whether some assumptions are oversimplified. Sharp transitions are very rare in natural systems (with exceptions of course, e.g., quorum sensing).

The magnification of the transition part of Fig. 2 (shown below) and Fig. 3c show that the loss of glucose and amino acid is not an abrupt 0 to 1 transition; instead, it is smooth but fast. This is due to

rapidly increasing bacterial abundance resulting from rapid bacterial growth and consequent high degradation of polymers. This point has now been clarified in the text (lines 246-255).

“The observed fast transitions of glucose and amino acids happen when the exponential growth of bacteria leads to high bacterial abundance and high degradation of polymers (Fig. 2, 3c). Similar fast changes in bacterial abundance and co-occurring resource utilization have been observed in an experiment showing degradation of transparent exopolymer particles derived from cultures of the coccolithophore *Emiliania huxleyi*⁴⁷. However, a very sharp transition has been observed for N₂ fixation rate (Fig. 2i, 3d). Because of the cost related to N₂ fixation, N₂ fixation starts only when there is no N available from amino acid (Fig. 3c). The rate of N₂ fixation is determined by the available electron acceptors (Fig. 3a) and results in a sharp transition at the beginning of N₂ fixation. N₂ fixation stops when the C available from glucose is not sufficient to burn excess O₂ to keep the cell O₂ free; consequently, N₂ fixation terminates abruptly.”

Growth rates and changes of bacterial numbers are simply impossible (3.6 d^{-1} ?). If the model needs growth rates that high to work, there is something wrong structurally. The change of bacterial numbers is much too fast for the decrease in bacterial numbers too (especially without predation). But are these abrupt changes in bacterial numbers even necessary for the model to work?

We respectfully disagree with the reviewer that the highest growth rates are impossible. As mentioned above, the observable occasional abrupt changes in bacterial abundances or substrate concentrations have been observed in natural systems. In fact, a number of studies report comparable rates, even in sinking particles. For instance, in an experiment on colonization, detachment, growth, and grazing mortality of microbial communities attached to model aggregates (4-mm-diameter agar spheres), Kjørboe et al. (2003) showed an experimental population change of 100 fold increase over 24 hours, which yields a growth rate estimate of $>4 \text{ day}^{-1}$. Datta et al. (2016) grew natural bacterial populations on artificial particles and found comparable growth rates (based on increase in RNA concentration). Friedrich et al. (1999) found 20-fold higher specific growth rates ($\sim 12 \text{ day}^{-1}$) for particle attached bacteria than their free-living counterparts. Smith et al. (1995) measured growth rates of bacteria attached to diatom aggregates of 4 to 16 day^{-1} .

This point is emphasized in lines 171-172: “The occurrence of such a high growth rate^{40,41} and bacterial abundance^{25,42,43} is not rare inside natural sinking particles.”

We would like to point out that predation is included in our model. The linear mortality term, that includes predation mortality, is represented by m_B and mentioned in the ‘Methods’ section as well as in Eq. 25(a) of Table 1. We agree that the term ‘predation’ was not explicitly mentioned before, but now we mentioned it in the modified ms (lines 592-593) :

“... and m_B represents the mortality rate (including predation) of bacteria”.

The narrative lacks a simple recap of the model, essentially a short stepwise explanation how the various factors lead to the outcome (perhaps a flow chart?). Beyond the mathematical outcome, does it make sense? The model has to work conceptually, therefore the authors should briefly report the causal sequence of events.

Thank you for this valuable point. We now provide a flow chart (Supplementary Fig. S3) explaining how different factors are responsible for different events inside particles, and we refer to this figure in the main text (lines 201-203).

“A stepwise conceptual flow chart of how different factors are responsible for different events inside particles is provided in Supplementary Fig. S3.”

We wish to note that we throughout the manuscript compare our mathematical outcome to empirical observations. For instance, cellular growth rates (lines 171-172), bacterial biomass (lines 171-172), anoxic particle interior (lines 174-176), and NO_3^- and SO_4^{2-} respiration (lines 181-182)). We also explained how cellular processes/conditions relate to particle dynamics; e.g. high cost of O_2 management during N_2 fixation (lines 240-242), early exhaustion of amino acids (lines 216-217), and the occurrence of an organic solute trail (lines 205-207). By relating the rather abstract mathematical outputs to empirical findings and observations, we hope that the results become digestible to most readers.

It would also be very useful for the reader to run a sensitivity analysis that shows which components most strongly affect the main outcomes (i.e., occurrence of nitrogen fixation inside the particle). This could be demonstrated by showing a matrix/table or heat map.

As suggested we have now performed a sensitivity analysis to examine how the N_2 fixation rate is affected by variations in the baseline parameters by $\pm 25\%$ (Supplementary Fig. S2). The procedure of the sensitivity analysis is described in the supplementary material (lines 127-139) and the main results are incorporated in the main text:

Main text (lines 185-193): “We performed a sensitivity analysis, adapted from earlier studies⁴⁵ to investigate how different factors affect the N₂ fixation rate inside particles (Supplementary Fig. S2). N₂ fixation rate was found to increase with increased maximum glucose uptake rate and maximum amino acid uptake rate, and decrease with the cost of amino acids uptake, hydrolysis rate of polysaccharide, and the fraction of O₂ diffusivity within particles compared to water. Significant decreases in N₂ fixation rate were observed with an increase in hydrolysis rate of polysaccharide and a decrease in maximum glucose uptake rate. Not surprisingly, the sensitivity analysis suggests hydrolysis and uptake as key parameters affecting N₂ fixation, pinpointing the importance of particle composition and bioavailability for particle-associated N₂ fixation.”

We also tested the sensitivity of N₂ fixation rates in sinking particles by varying the abundance and the proportion of large and small particles:

Main text (lines 388-392): “Since the abundance and size spectrum of particles vary with latitude and seasonally at high latitudes⁶⁴, we test the sensitivity of N₂ fixation rates by varying the parameter determining the abundance (n_0) and the proportion of large and small particles (ξ). We find that the N₂ fixation rate varies between 0.3-1.7 $\mu\text{mol N m}^{-3} \text{d}^{-1}$ inside sinking particles (Supplementary Fig. S6).”

Main text (lines 423-427): “The likelihood and rate of N₂ fixation associated with any individual particle will depend upon numerous factors. These include the vertical profiles of O₂ and NO₃⁻ through which the particle sinks, but in particular also the particle composition and bioavailability, including the initial polysaccharide and polypeptide concentrations, together with the size, sinking speed, and abundance of particles, as suggested by the model sensitivity analysis.”

Line 168 Replace seldom with rare

Revised as suggested.

References:

Datta, M. S., Sliwerska, E., Gore, J., Polz, M. & Cordero, O. X. Microbial interactions lead to rapid micro-scale successions on model marine particles. *Nat. Commun.* **7**, 11965 (2016).

Friedrich, U. Schallenberg, M. Holliger, C. (1999) Pelagic bacteria–particle interactions and community-specific growth rates in four lakes along a trophic gradient, *Microb Ecol*, **37**, pp. 49-61

Inomura, K., Bragg, J., Riemann, L. & Follows, M. J. A quantitative model of nitrogen fixation in the presence of ammonium. *PLoS One* **13**, e0208282 (2018).

Iversen & Jørgensen 1993, *Geochi Cosmochimica* **57**:571-578

Kjørboe, T., Tang, K., Grossart, H. P. & Ploug, H. Dynamics of microbial communities on marine snow aggregates: colonization, growth, detachment, and grazing mortality of attached bacteria. *Appl. Environ. Microbiol.* **69**, 3036–3047 (2003).

Medina LE, Taylor CD, Pachiadaki MG, Henríquez-Castillo C, Ulloa O, Edgcomb VP. **2017**. A review of protist grazing below the photic zone emphasizing studies of oxygen-depleted water columns and recent applications of in situ approaches. *Front. Mar. Sci.* 4: 105

Piontek, J (2009) Effects of temperature and pCO₂ on the degradation of organic matter in the ocean, PhD Thesis, University of Bremen, Bremen, Germany, page 121-164.

Ploug, H. & Passow, U. Direct measurement of diffusivity within diatom aggregates containing transparent exopolymer particles. *Limnol. Oceanogr.* **52**, 1–6 (2007).

Smith, D.C., Steward, G.F., Long, R.A., Azam, F. 1995. Bacterial mediation of carbon fluxes during a diatom bloom in a mesocosm. *Deep-Sea Res. II*; 1:75-97

Wright, J. & Colling, A. The Seawater Solution. in *Seawater: its Composition, Properties and Behaviour* 85–127 (El, 1995).

Reviewer #3:

This manuscript presents a mathematical model of nitrogen fixation as an optimal growth strategy utilized by heterotrophic bacteria in sinking marine particles. The model is well constructed and appropriate for the questions addressed. The analyses are insightful and clearly presented. The paper is well written and addresses important questions in microbial ecology and global biogeochemistry. The authors have done fine work and I think the manuscript can be published as is.

Thank you very much for recommending publication of our work.

REVIEWERS' COMMENTS

Reviewer #2 (Remarks to the Author):

The authors have sufficiently addressed my concerns throughout the manuscript and in the Supplementary section except that I still disagree with the authors on the growth rates. These high growth rates are far from being established at low temperatures, and more research is needed. Kiorboe et al. 2003 showed colonization rates of particles, not growth rates. This is something very different. Colonization means that motile bacteria actively seek particles in the water and attach to them.

Datta et al. conducted their experiments at room temperature which is very different from oceanic temperatures in the interior of the ocean modeled here.

I doubt the validity of growth rates based on leucine incorporation in mesocosm experiment with diatoms by Smith et al. 1995. They used a pore size of a 1 μm filter as a criterion of "attached bacteria". These rates could be caused by methodological problems with the leucine incorporation method. 1 μm is hardly in the size range of marine snow particles discussed in this paper.

Friedrichs et al. 1999 study was in lakes, at higher temperature, and using thymidine incorporation with possibly similar issues as with leucine incorporation.

Referring to the rates as "not rare" is misleading. I suggest that the authors qualify their statement: for instance, they could state that "the growth rates used in the model are high for the temperature regime but conceivable given that there are some reports of extremely high growth rates in particle-associated bacteria albeit at higher temperature and in different environments than modeled here (citations)."

The Kiorboe reference definitely needs to be removed.

REVIEWER COMMENTS

Reviewer #2:

The authors have sufficiently addressed my concerns throughout the manuscript and in the Supplementary section except that I still disagree with the authors on the growth rates. These high growth rates are far from being established at low temperatures, and more research is needed. Kiorboe et al. 2003 showed colonization rates of particles, not growth rates. This is something very different. Colonization means that motile bacteria actively seek particles in the water and attach to them.

Datta et al. conducted their experiments at room temperature which is very different from oceanic temperatures in the interior of the ocean modeled here.

I doubt the validity of growth rates based on leucine incorporation in mesocosm experiment with diatoms by Smith et al. 1995. They used a pore size of a 1 μm filter as a criterion of “attached bacteria”. These rates could be caused by methodological problems with the leucine incorporation method. 1 μm is hardly in the size range of marine snow particles discussed in this paper.

Friedrichs et al. 1999 study was in lakes, at higher temperature, and using thymidine incorporation with possibly similar issues as with leucine incorporation.

Referring to the rates as “not rare” is misleading. I suggest that the authors qualify their statement: for instance, they could state that “the growth rates used in the model are high for the temperature regime but conceivable given that there are some reports of extremely high growth rates in particle-associated bacteria albeit at higher temperature and in different environments than modeled here (citations).”

The Kiorboe reference definitely needs to be removed.

Thank you for accepting our revision on the previous version of the manuscript. Regarding the issue of high growth rate, we have removed the Kiorboe reference and modified the sentence according to the suggestion by the reviewer as (lines 181-184):

“The growth rates observed in the model are high for the temperature regime but conceivable given that there are some reports of extremely high growth rates in particle-associated bacteria albeit at higher temperatures and in different environments than modeled here^{40,41}”.